# Three open questions in polygenic score portability

Joyce Y. Wang[1], Neeka Lin[1], Michael Zietz [2], Jason Mares[3], Olivia S. Smith[1], Paul J. Rathouz[4,5] & Arbel Harpak [1,5] ✉

The broad adoption of polygenic scores (PGS) is hindered by their limited portability to people that differ—in genetic ancestry or other characteristics—from the GWAS samples used to construct them. Here, we measure PGS prediction accuracy as a continuous function of individuals' genome-wide genetic dissimilarity to the GWAS sample (genetic distance). Our results highlight three gaps in our understanding of PGS portability. First, variation in individual-level prediction accuracy is only weakly predicted by genetic distance. In fact, it is explained comparably well by socioeconomic measures. Second, trends of portability vary across traits. For several immunity-related traits, prediction accuracy drops near zero even at intermediate genetic distances—potentially reflecting fast evolutionary turnover of genetic variants associated with immunity. Third, even qualitative trends of portability can depend on how we measure predictive performance. For instance, for type 2 diabetes, precision remains roughly constant, while recall surprisingly increases with genetic distance. Together, our results show that portability cannot be understood through global ancestry groupings alone. Other, understudied factors influence portability, including the specifics of trait evolution, genetic architecture, social context, and the construction of the PGS. Addressing these gaps can aid in the development of PGS and inform more equitable applications.

Polygenic scores (PGS), genetic predictors of complex traits based on genome-wide association studies (GWAS), are gaining traction among researchers and practitioners[1–3]. Yet a major problem hindering their broad application is their highly variable performance across prediction samples[4–7]. Often, prediction accuracy appears to decline in groups unlike the GWAS sample—in genetic ancestry, social setting, environmental exposures, or other sample characteristics[4,6,8–11], restricting the contexts in which PGS can be used reliably.

This so-called portability problem is a subject of intense study. Typically, portability is evaluated through variation in the within-group phenotypic variance explained by a PGS (i.e., the coefficient of determination, $R^2$) among genetic ancestry groups. Indeed, population genetics theory gives clear predictions for the relationship between genetic dissimilarity to the GWAS sample and PGS prediction accuracy under some models (neutral evolution[12–14], directional[15], or stabilizing selection[13,15]), all else being equal (including, e.g., assumptions about environmental effects).

However, inference based on empirical variation in $R^2$ can be misleading for various reasons. For one, $R^2$ can be arbitrarily low even when the model fitted to the data is correct. It also cannot be compared across transformations of the data. $R^2$ is not comparable across datasets, because, for instance, it depends on the extent of variation in the independent variable[16–18]. In the context of inference about the causes of PGS portability, these issues can manifest in different ways[19].

[1]Department of Integrative Biology, The University of Texas at Austin, Austin, TX, USA. [2]Department of Biomedical Informatics, Columbia University, New York, NY, USA. [3]Department of Neurology, Columbia University, New York, NY, USA. [4]Department of Statistics and Data Science, The University of Texas at Austin, Austin, TX, USA. [5]Department of Population Health, The University of Texas at Austin, Austin, TX, USA. ✉e-mail: arbelharpak@utexas.edu

For example, heterogeneity in within-group genetic variance and environmental variance can each greatly affect group differences in $R^2$.

A related issue is that the impacts of environmental and social factors on portability are not well-understood, despite evidence illustrating these impacts can be substantial[4,6,19–21]. To complicate matters, such factors may be confounded with genetic ancestry, limiting our ability to make inferences based on the typical decay of $R^2$ between PGS and trait value in ancestries less represented in GWAS samples[3,4,6,20,22].

With these limitations of $R^2$, and the possible confounding with environmental and social factors, it remains unclear how well genetic ancestry would predict the applicability of PGS for individuals. Recent work implied that individual-level prediction accuracy should be largely explained by genome-wide genetic dissimilarity to the GWAS sample (see Fig. 3 in Ding et al.[23] and Fig. 5 in Tsuo et al.[24]). However, we note that this work focused on the relationship between genetic distance and the length of the prediction interval, i.e., expected uncertainty in prediction under a relationship with the prediction accuracy of the realized trait value. Understanding the drivers of variation in prediction accuracy is especially pertinent for personalized clinical risk predictions and decisions regarding their reporting to patients[2,25].

This motivated us to empirically study PGS prediction accuracy at the individual level. In what follows, we highlight three puzzling observations that also point to three gaps in our understanding of the portability problem: (1) Genetic dissimilarity to the GWAS sample poorly predicts portability at the individual level, (2) portability trends (with respect to genetic distance) can be trait-specific; and (3) portability trends depend on the measure of prediction accuracy. Informed by our results, we suggest avenues of future research that can help bridge these gaps.

## Results

### Portability and individual-level genetic distance from the GWAS sample

We examined PGS portability as a function of genetic distance from the GWAS sample in the UK Biobank (UKB). For each of the 15 continuous physiological traits, we performed a GWAS in a sample of 336,923 individuals. For 69,500 individuals not included in the GWAS sample (henceforth referred to as the prediction sample), we predicted the trait value using the PGS and covariates. Using a principal component analysis (PCA) of the genotype matrix of the entire sample, we quantify each individual's genetic distance from the GWAS sample as distance from the centroid of GWAS individuals' coordinates in PCA space (Fig. 1A). This measure is quicker to compute, yet highly correlated with $F_{st}$ between the GWAS sample and single individuals in the prediction sample (Pearson's $r > 0.9835$, $p$-value $< 2.2 \times 10^{-16}$), albeit noticeably less reflective of $F_{st}$ at intermediate genetic distances (Fig. 1B). The imperfect correlation may be a result of our use of only the top 40 PCs[12,26]. Under some theoretical conditions (such as neutral evolution, additive contribution of genotype and environment, fixed

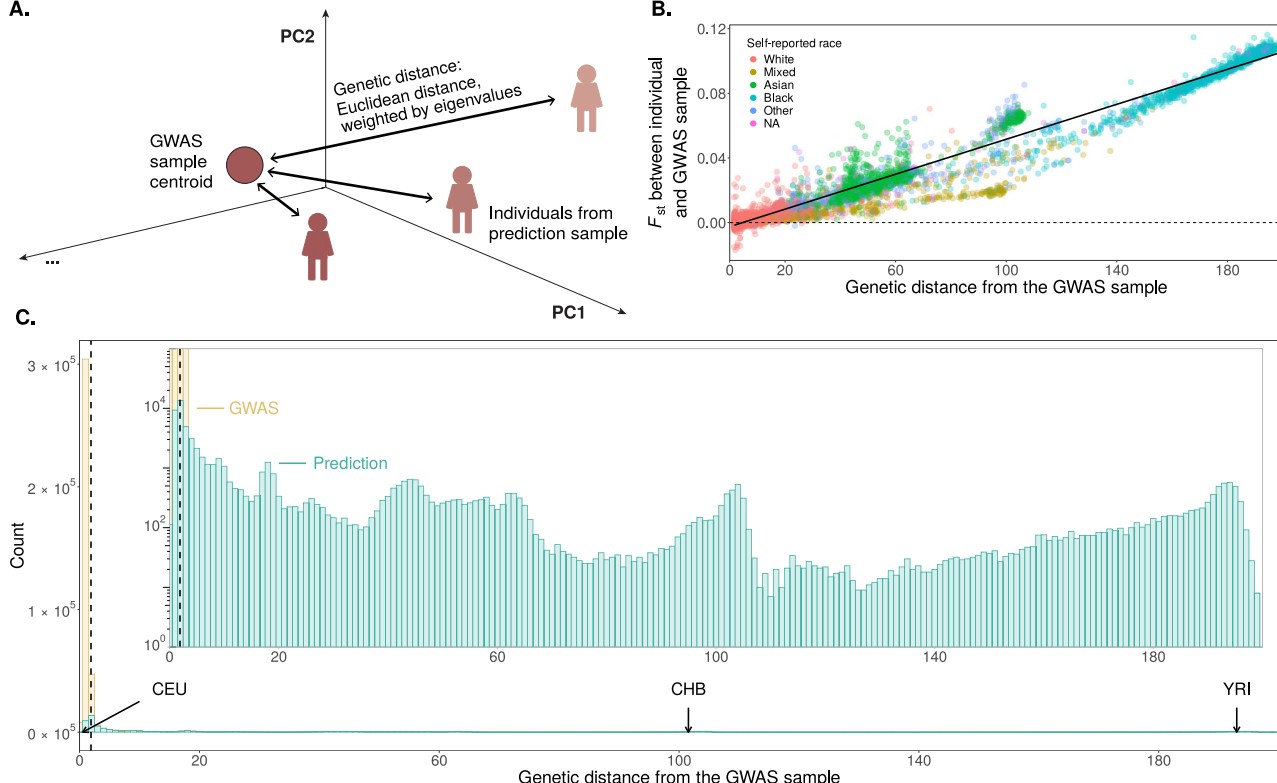

**Fig. 1 | Measuring genetic distance from the GWAS sample. A** Across 336,923 individuals in the GWAS sample and 69,500 individuals in the prediction set, we measure genetic distance from the GWAS sample as the weighted Euclidean distance from the centroid of GWAS individuals in PCA space, with each PC weight being proportional to its respective eigenvalue. **B** Across 10,000 individuals from the prediction set, genetic distance to the GWAS sample (calculated with 40 PCs) is highly correlated with $F_{st}$ between the GWAS sample and the individual (Pearson's $r > 0.9835$, $p$-value $< 2.2 \times 10^{-16}$; Supplementary Fig. 1). Under a theoretical model where portability is driven by genetic ancestry alone and the trait evolves neutrally, $F_{st}$ should perfectly predict variation in prediction accuracy. We note that genetic distance is less reflective of $F_{st}$ for intermediate genetic distances. **C** The distribution of genetic distance. For reference, we show the mean genetic distances for subsets of the 1000 Genomes Project Phase 3 dataset[28]: CEU, Utah residents of primarily Northern and Western European descent; CHB, Han Chinese in Beijing, China; YRI, Yoruba in Ibadan, Nigeria. The dashed line represents the 97.5th percentile of genetic distance from among GWAS sample individuals. In what follows, our reports are based on individuals with genetic distances larger than this value. The inset is a zoomed-in view of a smaller range and on a log scale, to better visualize the distribution within the prediction sample. For all panels, no repeated measurements from the same individual were taken.

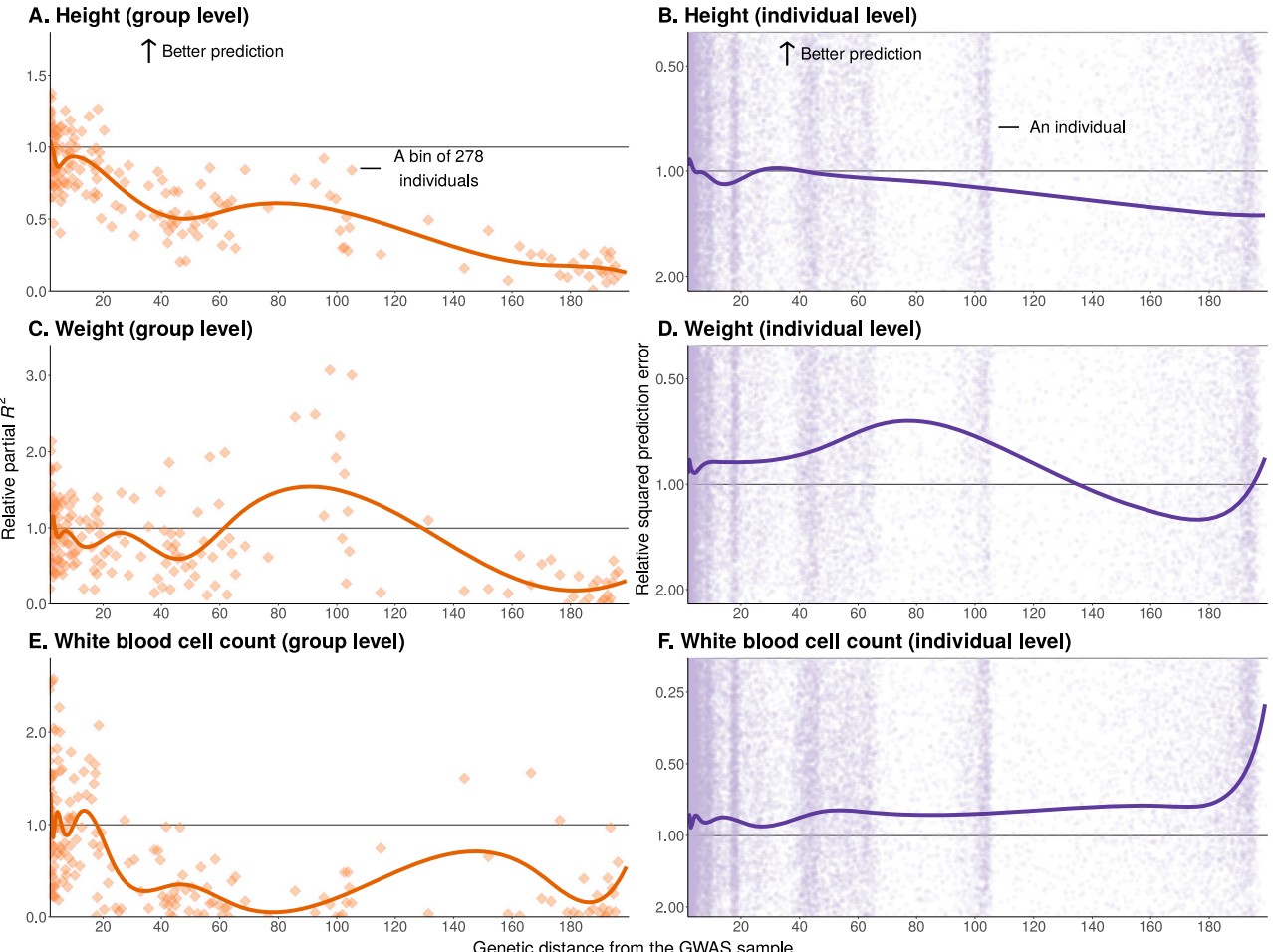

**Fig. 2 | Trends of portability vary across traits and measures.** At the group level (**A, C, E**), we measured prediction accuracy with the squared partial correlation between the PGS and the trait value in 250 bins of 278 individuals each. At the individual level (**B, D, F**), we measured prediction accuracy as the squared difference between the predicted phenotype and the true phenotype value. At both the group and individual levels, *y*-axis values show relative prediction accuracy, i.e., prediction accuracy divided by a baseline value. The baseline value is the mean prediction accuracy in 25 bins with median genetic distances that are closest to the mean genetic distance for GWAS individuals. Supplementary Table 1 details these baseline values for each trait. Curves show cubic spline fits, with 8 knots placed based on the density of data points. For height, prediction accuracy decays nearly monotonically with genetic distance at both the group (**A**) and individual (**B**) levels. **C, D** For weight, prediction accuracy does not monotonically decay with genetic distance. For white blood cell count, at the group level (**E**), prediction accuracy drops near zero at a short genetic distance from the GWAS sample; yet at the individual level (**F**), it increases. See Supplementary Figs. 2–5 for other traits and Supplementary Figs. 6–9 for plots showing the full ranges of individual-level prediction accuracy.

environmental variance)−$F_{st}$ should perfectly predict variation in prediction accuracy due to genetic ancestry[12,14,27]. We standardized genetic distance such that its mean is 1 across GWAS sample individuals.

In the prediction sample, we observed a continuum of genetic distance from the GWAS sample with several clear modes, the main one at short distances: 38,992 individuals have a genetic distance of up to 10, and the remaining 30,508 individuals at distances between 10 and 199.6 (Fig. 1B, C). To ground our expectations, we estimated the mean genetic distance for three 1000 Genomes[28] subsamples: Utah residents of primarily Northern and Western European descent (CEU) average at 0.5, Han Chinese in Beijing, China (CHB) average at 101.6, and Yoruba in Ibadan, Nigeria (YRI) average at 193.0 (Fig. 1C).

For each of the 15 continuous physiological traits, we measure the prediction accuracy at the group and individual level with slightly different prediction models ("Methods"). In both cases, we fitted a prediction model regressing the trait to the polygenic score and other covariates. To evaluate group-level accuracy, we split individuals into 250 bins of genetic distance, comprising of 278 individuals each. Within each bin we measure the partial $R^2$ of the polygenic score and the trait value. To evaluate individual-level accuracy, we measured the squared difference between the PGS-predicted value and the trait, after residualizing the trait for covariates.

### Prediction accuracy is weakly predicted by genetic distance

For some traits, such as height, group-level prediction accuracy decayed monotonically with genetic distance from the GWAS sample, as expected and reported previously (Fig. 2A)[12,23,29]. A major factor driving this decay appears to be an associated decay in heterozygosity in the PGS marker SNPs (single nucleotide polymorphisms) (Fig. 4B, Supplementary Fig. 22; see Patel et al.[15] and Wang et al.[29]). Lower heterozygosity in PGS markers impacts the genetic variance a polygenic score can capture because it makes for a less variable predictor. The impact of genetic distance on LD with causal variation is less straightforward[13,29,30].

Previous work implied that variation in individual-level prediction accuracy should be largely explained by genetic distance[23,24]. However, that was not the case in our analysis. While individual-level accuracy generally decayed with distance for most traits, this correlation was weak (Fig. 2B, and Supplementary Fig. 2). Even a flexible cubic spline fit of genetic distance explained little of the variance in prediction accuracy (e.g., $R^2 = 0.51\%$ for height).

**A. Mean trends in individual-level prediction accuracy**

**B. Deprivation index and genetic distance explain portability comparably well**

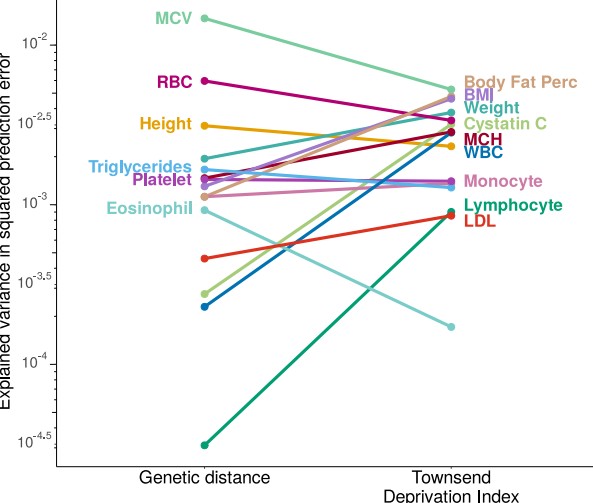

**C. Joint effect of genetic distance and SES on BMI prediction accuracy**

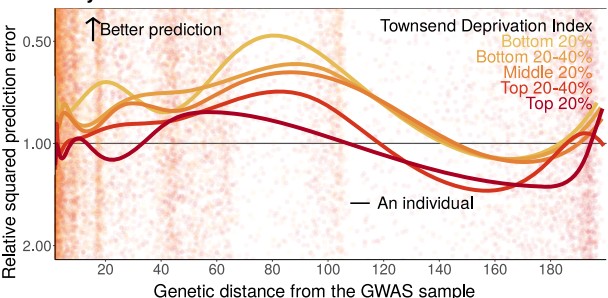

**Fig. 3 | Genetic distance and socioeconomic factors explain individual-level prediction accuracy comparably well. A** The *y*-axis corresponds to the mean (±SE) squared prediction errors of strata (subsets) of the 69,500 individuals in the prediction set. For genetic distance and Townsend Deprivation Index, individuals are binned into 5 equidistant strata, and the *x*-axis shows the median measure value for each stratum. Household income refers to the average yearly total household income before tax. The 5 strata for household income are taken directly from UKB data field 738. See Supplementary Figs. 14–17 for other traits. **B** We compared the variance in squared prediction error explained by a cubic spline fit to genetic distance to the variance explained by a cubic spline fit to the Townsend Deprivation Index in the prediction set. See Supplementary Figs. 18–21 for the variance explained by other genetic and socioeconomic measures. **C** We stratified the prediction sample into 5 equal-sized strata of Townsend Deprivation Index[31], and fitted

a cubic spline to squared prediction error within each stratum separately in the prediction set. PGS prediction accuracy is measured as the squared difference between the predicted and true PGS. The *y*-axis values show relative prediction accuracy, i.e., prediction accuracy divided by a baseline value. The baseline value is the mean prediction accuracy in 25 bins with median genetic distances that are closest to the mean genetic distance for GWAS individuals. The 5 curves show stratum-specific cubic spline fits, with 8 knots placed based on the density of data points. Top corresponds to the most deprived stratum. See Supplementary Figs. 67–70 for both group and individual levels for other traits. For all panels, no repeated measurements from the same individual were taken. MCV mean corpuscular volume, MCH mean corpuscular hemoglobin, RBC red blood cell count, Body fat perc, body fat percentage, BMI body mass index, WBC white blood cell count, LDL LDL cholesterol level.

In fact, individual-level prediction accuracy is explained comparably well by socioeconomic measures (Fig. 3B, and Supplementary Figs. 18–21). For example, we observed a steady mean increase in squared prediction error across quantiles of Townsend Deprivation Index[31] for 11 of the 15 traits examined, suggesting poorer prediction in individuals of lower socioeconomic status (Fig. 3A, C, and Supplementary Figs. 14, 16, 17; the four exceptions being white blood cell-related traits, Supplementary Fig. 15; see also similar reports in Mostafavi et al.[4], Hou et al.[6], and Nagpal and Gibson[21]). Like genetic distance, the Townsend Deprivation Index only explains between 0.02% and 0.53% of the variance in squared prediction error across traits with a cubic spline. Notably, however, for the majority of traits, more variance is explained by this measure of socioeconomic status than by genetic distance (Fig. 3B).

**Trends of portability vary across traits**

One might expect a qualitatively similar, monotonic relationship between genetic distance and prediction accuracy across traits.

Previous analyses (that have not examined individual-level prediction accuracy) observed similarly monotonic[8], and even linear[12,23], relationship regardless of the trait examined. However, we observed variation in this relationship among traits. Unlike the case of height, the prediction accuracy for many other traits did not decay monotonically with genetic distance. Weight and body fat percentage peaked in accuracy at intermediate genetic distances (Fig. 2D, and Supplementary Fig. 2).

In other traits we examined, in particular white blood cell-related traits, group-level prediction accuracy dropped near zero even at a short genetic distance (Fig. 2E, and Supplementary Fig. 3). There are multiple possible drivers of trait-specific portability trends. We considered, in particular, variable selective pressures on the immune system across time and geography. We hypothesized that these would lead to less portable genetic associations (across ancestry) compared to other traits. To test this prediction, we re-estimated the effects of index SNPs (SNPs included in the PGS, ascertained in the original GWAS sample) in two subsets of the prediction sample, one

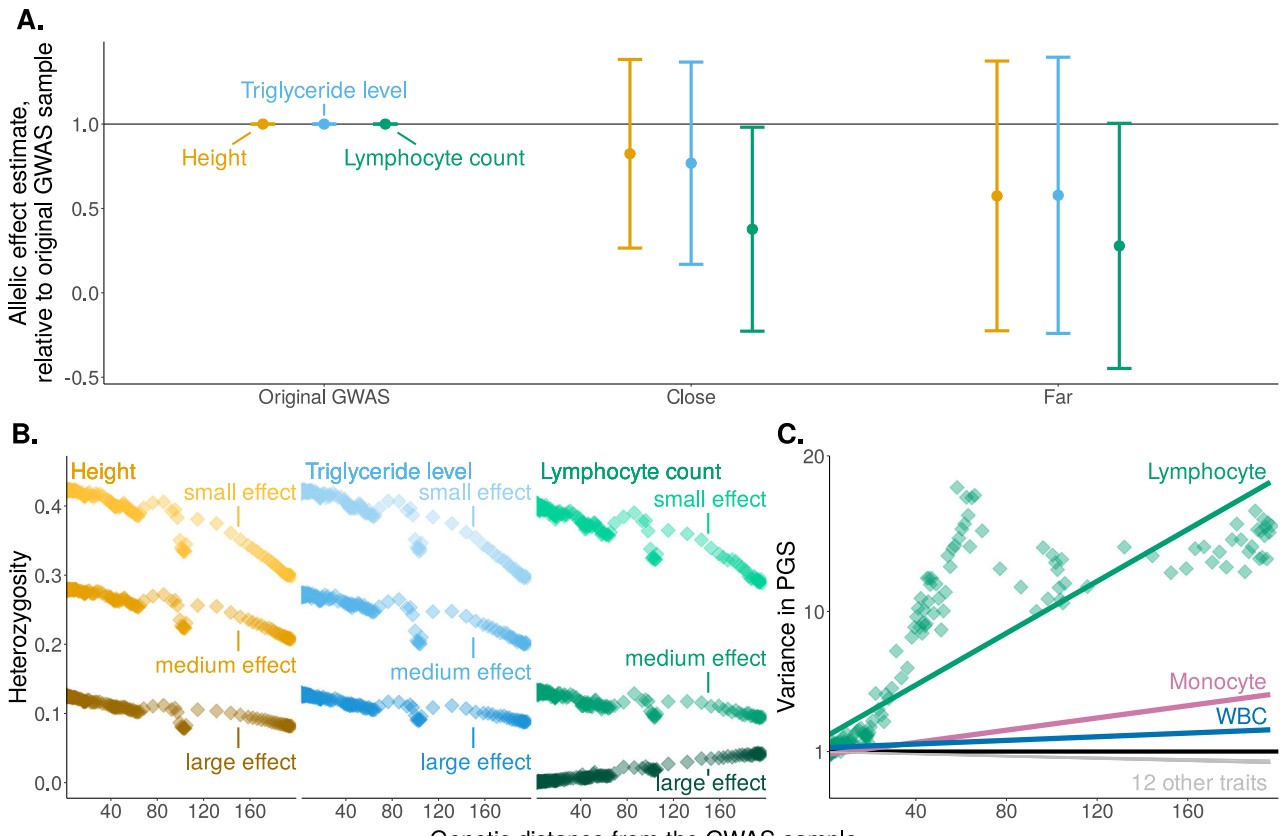

Fig. 4 | Lymphocyte count as an example of trait-dependent factors influencing portability. A We re-estimated the allelic effects of PGS index SNPs in subsamples of the prediction set: close (genetic distance ≤10, with 38,992 individuals), and far (genetic distance >10, with 30,508 individuals). For each index SNP of each PGS, we computed the allelic effect estimate relative to the effect estimate in the original GWAS sample (336,923 individuals). Shown are means ±standard deviations across PGS index SNPs for three traits, highlighting the poorer agreement between allelic effect estimates for lymphocyte count. B We compared the mean heterozygosity of index SNPs for height, triglycerides, and lymphocyte count. For each trait, SNPs are stratified into three equally-sized bins of squared allelic effect estimate (Supplementary Fig. 23). We also binned the prediction individuals into 250 bins of 278 individuals each by genetic distance. Each data point is the mean heterozygosity of a stratum in a bin of genetic distance. Unlike other traits, the heterozygosity of large

effect variants for lymphocyte count increases with genetic distance from the GWAS sample. See Supplementary Fig. 22 for other traits. C We binned the prediction individuals into 250 bins of 278 individuals each by genetic distance. We then compared the variance of PGS, in each bin, relative to the variance of PGS in the reference group, across traits. Among the 15 traits we have examined, only for lymphocyte count, monocyte count, and white blood cell count (WBC) the PGS variance increased with genetic distance. Green points show the PGS variance for lymphocyte count in genetic ancestry bins. Lines show the ordinary least squares linear fit to the respective bin-level data for each trait. The remaining 12 traits show similar and slightly negative slopes when relating PGS variance to genetic distance. For all panels, no repeated measurements from the same individual were taken.

closer and another farther (in terms of genetic distance) from the GWAS sample. The prediction sample-based allelic effect estimates were least consistent with the original GWAS for lymphocyte count, compared, e.g., to triglyceride levels, a trait of similar SNP heritability (Fig. 4A). To further illustrate this point, 31.7% of the index SNPs for lymphocyte count had a different sign when estimated in the original GWAS and in the closer GWAS, compared to 9.6% for triglyceride levels.

The rapid turnover of allelic effects may also interact with statistical biases. Consider, for example, the winner's curse, whereby effect estimates are inflated due to the ascertainment of index SNPs and the estimation of their effects in the same sample[32]. Winner's curse would be most severe in large effect PGS index SNPs: These SNPs are typically at lower frequencies in the GWAS sample than small effect index SNPs, because GWAS power scales with the product of squared allelic effect and heterozygosity[15,33,34]. If causal effects on lymphocyte count change rapidly, then large effect index SNPs may be under weaker selective constraint in the prediction sample than in the GWAS sample, and segregate at high allele frequencies. Indeed, for lymphocyte count, the heterozygosity of large-effect variants increases with genetic distance from the GWAS sample (Fig. 4B; see Supplementary Fig. 22 for other

traits). As a result of the trends of heterozygosity, the variance in the polygenic score (a sum over index SNP heterozygosity multiplied by their squared effect estimates) quickly increases with genetic distance for white blood cell count, lymphocyte count, and monocyte count, despite decreasing for all other traits we have examined (Fig. 4C, and Supplementary Fig. 66). And so, taken together, the PGS variance increases quickly and allelic effect estimates become non-predictive even close to the GWAS sample (Supplementary Fig. 24). Together, this may drive the immediate drop in prediction accuracy of white blood cell-related traits.

## The measure of predictive performance can alter our view of portability

Finally, the trends of portability, even qualitative trends, can depend on the measure of prediction accuracy. For triglyceride levels, lymphocyte count, and white blood cell count, group-level prediction accuracy is near zero far from the GWAS sample (Fig. 2, and Supplementary Figs. 3C and 5E) whereas at the individual level, prediction accuracy increases (Fig. 2F, and Supplementary Figs. 3D and 5E).

Trait-specific, measure-specific portability trends are of specific importance for diseases, since the choice of performance metrics

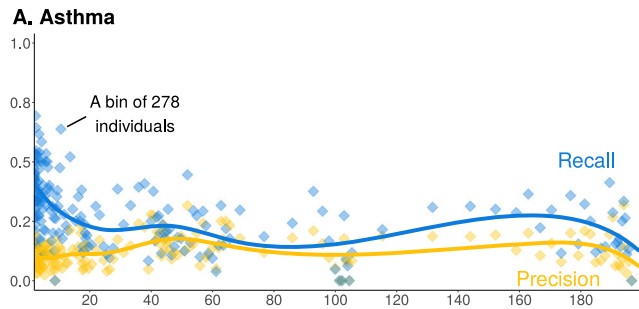

**A. Asthma**

A bin of 278 individuals

Recall

Precision

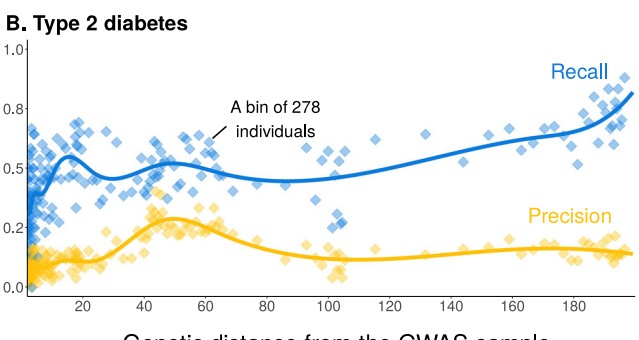

**B. Type 2 diabetes**

A bin of 278 individuals

Recall

Precision

Genetic distance from the GWAS sample

**Fig. 5 | Trends of portability vary across medical conditions and measures of predictive performance.** Individuals in the prediction sample were stratified into 250 bins by genetic distance. Each data point corresponds to a bin of 278 individuals. We calculated the precision (number of correctly-predicted cases divided by the number of predicted cases) and recall (number of correctly-predicted cases divided by the number of cases). **A** Asthma, with 5111 cases in the prediction sample. **B** Type 2 diabetes, with 5416 cases in the prediction sample. Curves show cubic spline fits, with 8 knots placed based on the density of data points. Note that in this figure, unlike Fig. 2, raw measures of prediction accuracy (precision and recall) are shown on the y-axis, rather than relative ones. For both panels, no repeated measurements from the same individual were taken.

often ties in with an intervention or policy that is being considered. For example, when a broadly accessible, safe intervention exists, we might prioritize the identification of all cases over all else and therefore focus on recall (number of true positives divided by number of true cases). In other cases, for example when the intervention includes a risky treatment, we might prioritize avoiding false positives and hence focus on precision (true positives over all positive classifications). Therefore, we sought to know whether trends of portability depend on the measures of predictive performance in disease.

To address this question, we analyzed precision and recall in two diseases with over 5000 cases in the prediction sample. For asthma, precision and recall depended on genetic distance in a qualitatively similar way (Fig. 5A). In contrast, for type 2 diabetes, precision appeared roughly constant for medium and large genetic distances, while recall generally increased with distance far from the GWAS sample (Fig. 5B). In conclusion, predictive performance showed trait-specific, measure-specific trends in disease risk prediction as well.

## Discussion

Through an examination of empirical trends of portability at the individual level, we highlighted three gaps in our current understanding of the portability problem. Below, we discuss possible avenues towards filling these gaps.

The driver of portability that has been extensively discussed in the literature is ancestral similarity to the GWAS sample[2,8,12,14,23,29]. Yet our results show that, at the individual level, prediction accuracy is poorly predicted by genome-wide genetic ancestry. We note that our measure of genetic distance (also similar to that used in other studies[6,12,23,35]) is

plausibly sub-optimal, as suggested, for example, by the noisiness of its relationship with $F_{st}$ at intermediate genetic distances (Fig. 1B). Therefore, one path forward is to ask whether refined measures of genetic distance from the GWAS sample, in particular ones that capture local ancestry[36,37] (e.g., in the genomic regions containing the PGS index SNPs), better explain portability. Another direction is in quantifying how environmental and social context, such as access to healthcare, affect portability (see Hou et al.[6] for a recent method in this vein). The relative importance of these factors will also inform the efforts to diversify participation in GWAS.

Second, we observed some trait-specific trends in portability, and hypothesize that they reflect the specifics of natural selection and evolutionary history of genetic variants affecting the trait. While previous work considered the impact of directional[14,15,38,39] and stabilizing selection[13,15,29,30] on portability, the trait-specific (and PGS-specific) impact—notably for disease prediction—is yet to be studied empirically. Here, we showed empirical support for the hypothesis that heterogeneous selective pressures on the immune system across time and geography may drive unique portability trends in immune-related PGS. Further research is needed to test this hypothesis and its generalizability. More generally, evolutionary perspectives on genetic architectures and other facets of GWAS data have been transformative[40,41]. This may also prove to be the case for understanding PGS portability.

Third, we show that individual-level measures, which are arguably the most relevant to eventual applications of PGS, can yield different results to group level measures that are widely used. PGS research has been focused on coefficient of determination ($R^2$) analyzed at the group level[4,8,12–14,23,29]. More generally, different applications and questions call for different measures of prediction accuracy, for instance when considering the utility of a public health intervention applied to communities, as opposed to asking about the cost-effectiveness of an expensive drug for an individual patient (see Abramowitz et al.[42] for related discussion). Therefore, future research on predictive performance could benefit from more focus on the metrics most relevant to the intended application.

Addressing these gaps in our understanding of PGS portability will be key for evaluating the utility of a PGS and for its equitable application in the clinic and beyond.

## Methods
### Data
**Data overview.** All analyses were conducted with data from the UKB, a large-scale biomedical database with a sample size of 502,490 individuals[43]. In this study, we considered 406,423 individuals who passed quality control (QC) checks, which included the removal of 359 individuals who withdrew from the study as of December 17, 2024, 651 samples identified by the UKB as having sex chromosome aneuploidy (data field 22019), and an additional 14,418 individuals whose self-reported biological sex (data field 31) differed from sex determined from that implied by their sex chromosome karyotype (data field 22001). We removed 963 individuals who are outliers in heterozygosity or genotype missingness (data field 22027) and 6851 individuals with genotype missingness greater than 2% (data field 22005). We then removed 72,825 individuals with 3rd-degree relatives or closer (data field 22020). In total, we removed 96,067 individuals. In the selection of the GWAS sample, individuals who passed all filtering steps and were labeled by the UKB as White British (WB)—those who self-identified as White and British, and closely clustered together in the PC space (336,923 individuals, data field 22006)—were included in the GWAS sample. The remaining 69,500 individuals who passed filtering (Non-White British, NWB) were used as the prediction set. In the Supplementary Materials, we investigate the sensitivity of our analysis to a different partitioning between the GWAS sample and prediction set.

**Genotype data.** We started with 765,067 biallelic variants out of a total of 784,256 genotyped variants on the autosomes. We first removed 10,543 SNPs within the major histocompatibility complex and extended region in strong LD with it (chromosome 6, positions 26,477,797–35,448,354 in the GRCh37 genome build). We excluded variants with a Hardy–Weinberg equilibrium $p$-value lower than $1 \times 10^{-10}$ among White British (WB) individuals (PLINK 2.0[44,45] flag `–hwe 1e-10`), removing another 45,930 variants. We also removed an additional 39,996 variants by setting the minor allele frequency threshold among WB to >0.01% (PLINK 2.0[44,45] flag `–maf 0.0001`). After filtering, we had 668,598 variants which we analyzed going forward.

**Phenotype data.** We analyzed 15 highly heritable continuous traits, as determined based on Neale Lab SNP heritability estimates[46] (Supplementary Table 1). These included both physiological measurements and biomarkers: standing height (data field 50), cystatin C level (data field 30720), platelet count (data field 30080), mean corpuscular volume (MCV, data field 30040), weight (data field 21002), mean corpuscular hemoglobin (MCH, data field 30050), body mass index (BMI, data field 21001), red blood cell count (RBC, data field 30010), body fat percentage (data field 23099), monocyte count (data field 30130), triglyceride level (data field 30870), lymphocyte count (data field 30120), white blood cell count (WBC, data field 30000), eosinophil count (data field 30150), and LDL cholesterol level (data field 30780).

In addition, we analyzed 2 binary traits, type 2 diabetes (ICD-10 code E11) and asthma (ICD-10 code J45). We marked an individual as positive for a disease if they are either positive for main ICD-10 (UKB field 41202) or secondary ICD-10 (UKB field 41204), and included all subtypes for both diseases. For all analyses, we only used the measurement from the first visit for each individual, and removed individuals with missing trait data.

## Genetic distance calculations

The fixation index ($F_{st}$) is a natural metric, a single number, to measure the divergence between two sets of chromosomes. We considered using it to measure the distance between the pair of chromosomes of an individual and chromosomes in the GWAS sample; however, calculating $F_{st}$ was computationally costly. Since previous work[12] showed it is tightly correlated with Euclidean distance in the PC space in the UKB, we used Euclidean distance as a single number proxying genetic distance from the GWAS sample. We used the pre-computed PCs provided by the UKB (data field 22009).

The genetic distance is the weighted PC distance between an individual coordinate vector $x$ in the subspace spanned by the first $K$ PCs and the centroid of the $M$ individuals $\{x^m\}_{m=1}^M$ in the GWAS sample, with $C = \frac{\sum_{m=1}^M x^m}{M}$, and is

$$\sqrt{\sum_{k=1}^K w_k (x_k - c_k)^2}$$

with weights

$$w_k = \frac{\lambda_k}{\sum_{n=1}^{40} \lambda_n},$$

where $\lambda_k$ is the $k$th eigenvalue.

To identify $K$, the number of PCs we used, and to confirm the approximation is reasonable for our data, we examined the correlation of genetic distance with $F_{st}$ as a function of $K$ on a small subset of the prediction sample.

We randomly selected 10,000 prediction sample individuals with a weighted PC distance greater than the weighted PC distance of 95% of the GWAS set (based on weighted PC distance calculated using $K = 10$). For those individuals, we estimated their $F_{st}$ and weighted PC distance to the GWAS centroid for $K \in 1,..., 40$ (UKB provides pre-computed individual-level coordinates for each of 40 PCs). We estimated $F_{st}$ in this subsample with the Weir and Cockerham method[47] using the `–fst` flag in PLINK 1.9[45,48].

Since the PC distance calculated from using $K = 40$ correlated most strongly with $F_{st}$ (Pearson's $r > 0.9835$, $p$-value $< 2.2 \times 10^{-16}$) (Supplementary Fig. 1), we used this number of PC to estimate the genetic distance for all test individuals (Fig. 1B, C). We note that genetic distance is less reflective of $F_{st}$ for intermediate genetic distances (Fig. 1B).

We divided the raw genetic distances by the (raw) mean genetic distance among GWAS sample individuals. To gain intuition about these standardized units of genetic distance, we wished to estimate where on this scale we would find individuals from three subsamples from the 1000 Genomes Project Phase 3 dataset[28]: CEU, Utah residents (CEPH) from primarily Northern and Western European descent; CHB, Han Chinese in Beijing, China; and YRI, Yoruba in Ibadan, Nigeria. To this end, we ran a PCA with a dataset that includes both the UKB individuals and the CEU, CHB, and YRI individuals. We identified the UKB individuals with the shortest weighted Euclidean distance to the centroid of each of the three 1000 Genomes populations, and used the genetic distance of those three UKB individuals in our PCA of only UKB individuals as a proxy of where the three 1000 Genomes subsamples fall on the scale of our genetic distance measurement (Fig. 1C).

The distribution of genetic distance is heavily right-skewed, with most individuals falling close to the GWAS centroid. Since we wanted to focus on the individuals far away from the GWAS set, we only analyzed data for individuals with a genetic distance greater than the 97.5th percentile of genetic distance from among GWAS sample individuals (Fig. 1C), with the exception of the analyses behind Fig. 3 and Supplementary Figs. 14–21.

For group level analyses, we binned the prediction samples by genetic distance using 250 equally-sized bins, with 278 individuals per bin.

## PGS and evaluating PGS prediction accuracy

**GWAS.** For each trait, we used the `–glm` flag from PLINK 2.0[44,45] to run GWAS on the GWAS set. We used the following covariates: the first 20 PCs from UKB (data field 22009), age (data field 21022), age², sex (data field 31), age × sex, and age² × sex (× denotes the product of two variables, conferring to an interaction term).

Using PLINK 2.0[44,45] with the flag `–glm` along with `–1`, we also ran a GWAS for 2 binary disease traits: asthma (ICD-10 code J45) and type 2 diabetes (ICD-10 code E11), using the same covariates as the ones used for continuous traits.

For both continuous and disease traits, we clumped SNPs using the `–clump` flag in PLINK 1.9[45,48], setting the association $p$-value threshold for clumping to 0.01 (`–clump-p1 0.01`), LD $r^2$ threshold to 0.2 (`–clump-r2 0.2`), and window size to 250 kb (`–clump-kb 250`).

**PGS construction.** After clumping and thresholding the SNPs with marginal association $p < 1 \times 10^{-5}$, we calculated PGS for each individual for every phenotype. The calculations were carried out with the `–score` flag in PLINK 2.0[44,45].

**PGS prediction accuracy at the group level.** To evaluate prediction accuracy for continuous trait PGS at the group level, we linearly regressed the phenotype on the covariates (array type (data field 22000), age, age², sex, age × sex, and age² × sex) and PGS within each genetic distance bin (phenotype ~ covariates + PGS), which is the full model. These analyses rely on standard linear-model assumptions, including linearity of effects, independence of observations, and approximately normally distributed residuals. We then performed another linear regression of the phenotype on the covariates,

excluding the PGS, within each bin (phenotype ~ covariates), which is the reduced model. Using these two squared correlations, we calculated partial $R^2$ for the PGS with the sum of squared errors of these two models as

$$R^2_{\text{partial}} = \frac{\text{SSE (reduced model)} - \text{SSE (full model)}}{\text{SSE (reduced model)}},$$

which represents the prediction accuracy of PGS for each bin.

As a baseline prediction accuracy, we identified the 25 bins (of 278 individuals each; 10% of the bins) with the median genetic distance most similar to the mean genetic distance for GWAS individuals; this reference group represents individuals from the prediction set that are closest to the typical GWAS individuals in terms of genetic distance. The mean PGS prediction accuracy across these 25 bins served as the baseline value. Throughout the paper, we report the prediction accuracy at the group level as a bin's squared partial correlation between the PGS and the trait divided by this baseline value.

For binary traits, we calculated the precision (fraction of true cases out of predicted cases) and the recall (fraction of predicted cases out of all cases) in each genetic distance bin across 250 bins of 278 individuals each. To get the predicted phenotype, we used the PGS and predicted the phenotype to be positive if their PGS is in the 75th percentile of the PGS of the GWAS set for type 2 diabetes disease and 70th percentile for asthma. These thresholds were determined based on the percentile (considering 5% increments) cutoff which maximized the F1 score in the GWAS set for each polygenic score.

$F_1$ score is defined as

$$F_1 = \frac{2 \times \text{precision} \times \text{recall}}{\text{precision} + \text{recall}},$$

where precision is defined as

$$\text{precision} = \frac{\text{TP}}{\text{TP} + \text{FP}},$$

and recall is defined as

$$\text{recall} = \frac{\text{TP}}{\text{TP} + \text{FN}},$$

where TP is true positives, FP is false positives, and FN is false negatives. Using this procedure, the optimal thresholds for asthma and type 2 diabetes were the 70th and 75th percentiles, respectively, as noted above.

**Prediction error at the individual level.** For the individual-level prediction error, we first derived phenotypic values adjusted for covariates $Z$ in two steps, involving residualizing some covariates in each genetic ancestry bin independently and some covariates globally. First, we regress raw phenotype values $Y$ independently in each bin on covariates, $Y \sim$ array type + age$^2$ + sex + age × sex + age$^2$ × sex. In bins in which only a single individual was genotyped with a particular array type, we did not include array type as a covariate. We then regress the residual $X$ (where $X = Y - \hat{Y}$ and $\hat{Y}$ is the fitted value from the first step) globally on covariates, $X \sim$ genetic distance polynomial + sex + sex × genetic distance, where genetic distance polynomial is a 20-degree polynomial in genetic distance. Finally, we regress the residual of this second regression, $Z = X - \hat{X}$ onto the PGS in a simple (univariate) linear regression. We refer to the squared residual of this regression,

$$\left(Z - \hat{Z}\right)^2,$$

as the unstandardized squared prediction error. Similar to the group-level analysis, we computed the mean unstandardized squared prediction error in the 25 reference bins as a baseline value (Supplementary Table 1 details the baseline values across traits). The squared prediction error, the measure of individual-level prediction accuracy we refer to throughout, is the unstandardized squared prediction error divided by the baseline value.

**Spline fits.** For both the individual-level and group-level analysis, Fig. 2 and Supplementary Figs. 2–9 show cubic spline fits. We fitted these splines using 8 knots with `bs()` from the R library `splines`[49]. The knot positions were chosen based on the density of the individual genetic distances, such that there is an equal number of samples between any two knots. This resulted in knots at genetic distances of 2.39, 3.54, 5.99, 11.20, 23.75, 45.40, 65.49, and 167.53.

**Mean trends in individual-level prediction accuracy.** In the Results section of the main text, we discuss various individual predictors of squared prediction error. In addition to genetic distance, we considered 2 measures of individual-level prediction error: The Townsend Deprivation Index[31] (data field 189) and average yearly total household income before tax (data field 738). For genetic distance and the Townsend Deprivation Index, we considered five uniformly-spaced bins, and computed the mean squared prediction error and the standard error of this mean (Fig. 3A, and Supplementary Figs. 14–17). For household income, which the UKB provides as categorical data conferring to ranges in British Pounds (£), we converted the categories into an ordinal variable coded as 1, 2, 3, 4, and 5, and computed the mean squared prediction error, and standard error of the mean in each. This also allowed us to use the income categories directly as measures in the regression models used for comparison of variance in prediction error explained that we discuss below.

**Comparison of variance in prediction accuracy explained across measures.** For this analysis, we used all the individuals in the prediction set and did not filter for the individuals with a genetic distance greater than 97.5th percentile of genetic distance from among GWAS sample individuals. We compared the variance in squared prediction error explained for 8 raw measures: genetic distance, Townsend Deprivation Index[31] (data field 189), average yearly total household income before tax (data field 738), educational attainment (data field 6138; which we converted into years of education according to Table S1 in Carter et al.[50]), minor allele counts for SNPs with different magnitudes of effects (three equally-sized bins of small, medium, and large squared effect sizes, see Supplementary Fig. 23), and minor allele counts of all SNPs. Minor here is with respect to the GWAS sample, and the count is the total sum of minor alleles across index SNPs of the magnitude category. Namely, for each measure, we independently fit three different models: (1) a linear predictor, fitted using ordinary least squares; (2) a discretized predictor, using one predicted value for each of the 5 bins where all five bins had identical widths; and (3) a cubic spline, with 16 knots placed based on the density of data points at the individual level, such that there was an equal number of data points between each pair of consecutive knots. After fitting the models, we calculated the $R^2$ values to determine the variance explained by each measure-method combination. We then computed 95% central confidence intervals for these $R^2$ values to assess the reliability of the estimates (Fig. 3B, and Supplementary Figs. 18–21).

**Additional analyses on lymphocyte count**
**Comparing allelic estimates across three GWASs.** To test whether allelic effect estimates are similar across genetic ancestry, we performed two additional GWASs for each trait in two subsets of the prediction sample: close group (genetic distance ≤10, with 96,457 individuals) and far (genetic distance > 10, with 32,822 individuals). For

both groups, we adjusted for 20 PCs of the genotype matrix of the respective set of individuals, using the `-pca approx 20` flag in PLINK 2.0[44,45]. After running GWAS independently in the two groups, for each index SNP of the original PGS, we divided allelic effect estimates in the original GWAS, close, and far set by the allelic effect estimate in the original GWAS. Figure 4A shows the mean ± standard deviation across PGS index SNPs for each of three traits.

**Heterozygosity at index SNPs as a function of genetic distance.** For each PGS, we calculated the heterozygosity of each index SNP in each bin from allele counts using the `-freq` flag from PLINK 1.9[45,48]. We stratified index SNPs into three equally-sized bins based on their squared effect sizes (Supplementary Fig. 23). Figure 4B and Supplementary Fig. 22 show the mean heterozygosity (across stratum SNPs) for each stratum in each genetic distance bin.

**Variance of PGS as a function of genetic distance.** For each phenotype, we calculated the variance of PGS in each bin relative to the mean of the variance of PGS in the 25 bins close to the GWAS set. In Fig. 4C, we plotted the values in each bin as well as a linear fit for lymphocyte count. For other traits, we only plotted the linear fit.

**Heritability associated with each index SNP.** We estimated the heritability explained by each index SNP as

$$2p(1-p)\widehat{\beta}^2,$$

where $\widehat{\beta}$ is the estimated allelic effect and $p$ is the allele frequency. In Supplementary Fig. 24, we compared the distribution of index SNP heritability across traits and with allelic effect estimates and heterozygosities calculated both in the original GWAS sample, the close prediction sample and the far prediction sample. For each trait, the SNPs used are also stratified into three equal-sized strata (small, medium, and large) based on their squared effect sizes, as discussed above.

**Reporting summary**

Further information on research design is available in the Nature Portfolio Reporting Summary linked to this article.

## Data availability

All data used for this study were obtained from the UK Biobank under application 61666. Instructions for access are available at https://www.ukbiobank.ac.uk/use-our-data. GWAS summary statistics are available at https://www.harpaklab.com/data. The 1000 Genomes Project Phase 3 dataset was used as an external reference to illustrate genetic distances among populations and can be accessed at https://www.internationalgenome.org.

## Code availability

The scripts used for the various analyses presented are available on GitHub[51] at https://github.com/harpak-lab/Portability_Questions and archived on Zenodo at https://doi.org/10.5281/zenodo.17921472.

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

## Acknowledgements

We thank Ipsita Agarwal, Maryn Carlson, Yi Ding, Doc Edge, Kangcheng Hou, Hakhamanesh Mostafavi, Bogdan Pasaniuc, Molly Przeworski, Sam Smith, Jeff Spence and members of the Harpak Lab for helpful feedback. This work was funded by NIH grant R35GM151108, a fellowship from the Simons Foundation's Society of Fellows (#633313) and a Pew Scholarship to A.H. This study was conducted using the UKB resource under application 61666, which received ethics approval from the North West Multi-centre Research Ethics Committee (REC reference number 11/NW/0382), and all participants provided written informed consent to UKB. This study was approved by the University of Texas at Austin institutional review board (protocol 2019-02-0125). We acknowledge the Texas Advanced Computing Center (TACC) at The University of Texas at Austin for providing computational resources that have contributed to the research results reported within this paper.

## Author contributions

J.Y.W. and A.H. wrote the paper with input from all co-authors. J.Y.W., N.L., M.Z., J.M., and O.S.S. performed analyses. P.J.R. contributed to methodological development. A.H. supervised the work.

## Competing interests

The authors declare no competing interests.
