## [Transparent Peer Review file · Nature Communications]

Three Open Questions in Polygenic Score Portability

Corresponding Author: Dr Arbel Harpak

Version 0:

Reviewer comments:

Reviewer #1

(Remarks to the Author)

Wang and colleagues use the UK Biobank to highlight what they call three open questions that remain to be understood about the (poor) portability of polygenic scores (PGS). They do not provide answers to any of these three remaining questions and leave this for future work. I am not sure what to get from this paper. I also have concerns about the validity of some of the analyses.

Main comments:

- "Previous reports suggested that the relationship between genetic distance and prediction accuracy is similar across traits [17,27,7]". I don't think this is true; these papers present the trend averaged across traits, but also acknowledge differences across traits (e.g. Fig 4 of Ref 7 and Fig 2 of Ref 27).
- At least one of the open questions that the authors mention is why is there some PGS portability issue and why is it different across traits? The consensus on the main reason behind the poor portability of PGS is that we use tagging variants in the PGS (instead of causal variants). These tagging variants then can have different allele frequencies, and especially different LD with the causal variants across ancestries, which can make them bad proxies of causal variants in other ancestries, resulting in a decrease in PGS performance. I don't think this is mentioned in this paper; instead rather vague possible explanations are mentioned, which I think don't help move the field forward.
- I have many concerns on how PGS and genetic distances were derived in this paper.
 - 1/ PGS are derived using the C+T method with $p < 1e-5$. This will likely use a very restricted number of variants, which could result in non-normal PGS distribution, which could cause R^2 to be biased or at least highly variable. Also, I don't understand how the authors can derive individual PGS intervals or accuracies when not using Bayesian sampling of causal effects (as done in Refs 6 & 7).
 - 2/ In Ref 27, genetic distance is directly computed from the (squared) Euclidean distance of PCs (which are already weighted by eigenvalues, so I do not think these should be weighted more).
 - 3/ Partial- R^2 values between 0 and 4 are odd. Use the true definition of a partial correlation: the correlation between the PGS and the outcome, after they have been both residualized by covariates. You can use e.g. `bigstatsr::pcor()` in R. Why not use the deprivation index in the adjusted covariates? Also, I don't think it is valid to compute an R^2 in possibly very heterogeneous groups (the same bin on genetic distance can include very different genetic ancestries). If you want to use more groups than in Ref 27, have a look at the many homogeneous non-UK groups defined in <https://doi.org/10.1093/bioinformatics/btac348>.
 - 4/ Use only 16 PCs from the UK Biobank; PC19-40 capture only LD and should never be used (cf. <https://doi.org/10.1093/bioinformatics/btaa520>).
 - 5/ I don't think there has been any filtering on relatedness in these analyses, which can bias predictions for related individuals.

Additional comments:

- Not sure it is valid to compute F_{st} between one individual and a group.
- Discussion materials should be more separated from the Results.
- "We standardized genetic distance such that its mean is 1 across GWAS sample individuals." -> what is the rationale behind this? Instead, you could scale using the max distance and max F_{st} , as done in Ref 27.

Signed Florian Privé

(Remarks on code availability)

Reviewer #2

(Remarks to the Author)

Review for "Three Open Questions in Polygenic Score Portability"

Joyce Y. Wang et al described three open questions for PGS analysis, including high variability, trait-specific portability and measurement impact. Overall, the study highlights that understanding PGS portability requires considering more than just global ancestry, including trait evolution, genetic architecture, social factors, and PGS construction. This approach could lead to more effective and equitable genomic research.

The study topics are very interesting. Manuscripts are well-written with clear illustration. Detailed major comments can be found in the following section.

Major:

1. A "table 1" should be included to describing the distribution of data set. Since authors examined variables like Townsend Deprivation Index, Household Income etc, it would be necessary to compare between GWAS sample and PGS sample to make sure there is no significant differences between the group. Besides variables used in the analysis, some other variables should be included in table 1 include age, sex, location (England, Scotland or Wales).
2. Line 99-100 & Section Prediction accuracy is weakly predicted by genetic distance & Figure 2: Authors split samples into a bin of approximately 260 individuals and compared the prediction accuracy against the genetic distance from the GWAS sample. (1) In Figure 2B regarding weight, the smoothed curve around 100 on x-axis seems to be influenced by a few points on the top. Some further analysis could be done to rule out the randomness of the sample split. For example, would that be possible to vary the number of bins (currently 500) to see if the current trend would maintain. (2) Additionally, currently analysis uses cubic spline fits with 8 knots to capture the trajectory of the trend. It would be helpful to fit some models, like GAM (generalized additive model) to show if the trend is significant or not. And adding the CI (confidence interval) around the smoothed line. (3) Figure 2 500 bins plot with individual ones are not quite corresponding to each other. The overall trend between the 2 are not consistent. For example, E and F, the whole trajectory in E is below 1. However, in F the line is above 1.
3. Line 183-189 in Discussions: Authors observed some trait-specific trends in portability. It would be great to have some additional analysis to examine what groups of traits might need additional inspections. Expanding on the potential implications of these findings for equitable genomic research could strengthen the conclusion.
4. Line 287-297: Authors detailed described how GWAS are constructed within UKB. (1) Did authors also take sample relatedness into consideration? Either model the sample relatedness or excluded those individuals that are closely related. These individuals might influence GWAS performance. Some more recent software, like REGENIE would be a better approach for modeling GWAS. (2) UKB data is collected at 3 regions, ie England, Scotland and Wales, in multiple centers in different batch. It would be great to also control assessment center (field id 54) and genotype array (field 22000) in the analysis.
5. Line 298-300: Authors use clumping and thresholding method to construct PGS model. Would that be possible to utilize other methods, like LDpred2 and PRSCS to examine the performance of PGS? Literatures have showed that these methods show a better performance comparing to clumping and thresholding strategy.

Minor:

1. Figure 1C: would that be possible to have a similar histogram for UKB samples?
2. It would be interesting to examine relationship between genetic PCs and F_{st} estimated in the study.
3. Line 171-176 & Discussions: It would be great to examine some admixed ancestry population performance considering the mosaic genetic structure. F_{st} is still a measure of genetic distance in a global level.
4. Line 340: field should be 22189 instead of 189.

In all, the manuscript described important questions in PRS analysis that GWAS summary statistics have different characteristics, especially ancestry with individual level data. While the manuscript is well written and ideas are new, some detailed information need to be refined.

(Remarks on code availability)

Reviewer #3

(Remarks to the Author)

(Remarks on code availability)

Scripts are well-written and easy to replicate.

Reviewer #4

(Remarks to the Author)

Wang et al. highlight three gaps for PGS portability with comprehensive analysis in the UKB. But I still have some concerns.

1. You only used 15 highly heritable continuous traits to study PGS's portability. To my knowledge, the selected traits are polygenic. I assume that the genetic architecture may be a contributing factor to PGS portability. I would like you to add the binary traits, such as type II diabetes (T2D) and Alzheimer's disease (AD). For example, the genetic architecture of AD might be a mixture of polygenic and sparse. Both the difference between continuous trait and binary trait and the difference between genetic architecture might result in different trends between PRS and genetic distance.
2. You only used a simple method to construct PGS, which might result in a low prediction performance. I recommend incorporating a PGS method into the sensitivity analysis. Based on your pipeline, you can use polygenic methods without a validation set, such as SBLUP and DSBLMM. It's intriguing to see if a better PGS yields a different trend for the same trait.
3. Based on the partial R2 equation, I assume that the value should be lower than 1. I'm confused as to why some groups' partial R2 of weight, BMI, and body fat percentage is greater than 1. I also suggest that you provide the raw R2 to demonstrate PGS's performance.
4. The variance of PGS for WBC, lymphocyte count, and monocyte count increases with genetic distance. Please conduct further analysis to delve deeper into the trend, particularly examining the specific genetic architecture of immunity-related traits. If you add asthma, an immunity-related trait, the trend should be similar.
5. You investigate the heterozygosity and SNP effect size in different traits. I assume that the LD structure is also a factor in PGS portability. Please perform additional analysis of LD structure, such as LD score.

(Remarks on code availability)

Version 1:

Reviewer comments:

Reviewer #1

(Remarks to the Author)

I am still unconvinced by this paper and have several comments.

Main comments

- Gap 1 & 3: Results (trends of predictive performance vs genetic distance to training) using the prediction error at the individual level do not agree with the ones at the group level. This is very confusing (I guess this is the authors' point). The problem I see is that the squared difference in predicted vs true value depends heavily on things like phenotypic variance and also random shifts in PGS distribution (which are corrected for when using some proper partial correlation at the group level).
- To get better quality results, the authors should present results
 - + with PRS-CS instead of C+T everywhere given the higher number of variants used with make the results less noisy (C+T could simply pick a few highly-transferable genetic variants by chance)
 - + with GAM instead of the cubic splines, which is much more robust and informative; you can also consider log-distances to have more evenly-spaced bins
 - + the authors should really consider log-transforming all the phenotypes they used (before residualizing) as they are probably all log-normally distributed (even height to a lesser extent -> just have a look at the distributions of males and females); this would make their result more robust to the long right tails of the phenotypic distributions.
 - + consider whether the individual-level predictions are just simply shifted due to differences in allele frequencies, which is corrected when adjusting PGS for PCs (cf. first main comment).
- Gap 2: "portability trends (with respect to genetic distance) can be trait-specific" -> again, this has been shown in previous papers and I don't consider this as a knowledge gap. Moreover, I think the use of C+T with just a few variants in the PGS make them even more trait-specific simply by chance of picking +/- transferable genetic variants, which I consider misleading (therefore the recommendation of using a more genome-wide PGS method like PRS-CS).

Additional comments:

- The figures in the response to reviewers were not of particularly good quality.
- Ding et al. have looked at individual-level prediction accuracy; that was the whole point of the paper. They used some estimates, but still.
- "The previous results on (population genetic) theory-based expectations for portability are discussed explicitly and referenced thoroughly." -> which still says nothing on LD and AF differences..

(Remarks on code availability)

Reviewer #2

(Remarks to the Author)

Joyce Y. Wang et al described three open questions for PGS analysis, including high variability, trait-specific portability and measurement impact. The manuscript had a great improvement since the last version. Majority of previous issues that were raised last time are properly addressed. Various sensitivity analyses validated the robustness of analysis conclusions. Here are some additional major and minor suggestions for authors to consider:

Some major changes could be:

1. Line 76, fit of GWAS and PRS analysis: authors subsetted 336923 individuals for GWAS analysis and applied to prediction sample for PRS. Notably, the GWAS was conducted exclusively in White British individuals, while the PRS was evaluated in a non-White British population. Given the known impact of population structure and genetic ancestry on PRS transferability, this discrepancy raises concerns about the validity of cross-ancestry comparisons. It would strengthen the analysis and interpretation if the authors could also include a subset of White British individuals in the PRS prediction phase. Comparing PRS performance between White British and non-White British individuals—using the same GWAS-derived weights—would provide a more rigorous assessment of ancestry-related biases and enhance the robustness of the conclusions. In that case, including the White-to-White group as a baseline would serve as a fundamental point of comparison.

2. Currently, all analyses—including both GWAS and PRS—are conducted within the UK Biobank. To strengthen the generalizability of the findings, it would be valuable to assess external validity using independent cohorts. For example, applying GWAS results from an external dataset to train PRS and testing them in the UK Biobank, or vice versa, could help rule out potential UK Biobank-specific cohort effects and enhance the robustness of the conclusions. Some potential large biobank can be used include: All of Us etc.

Minor issues:

1. Line 76-78: please explain how the GWAS sample and prediction sample divided.
2. Please keep tense consistency through the manuscript. Some places present tense (e.g. line 114) while some locations use past tense (e.g. line 113).

In summary, the manuscript shows significant improvement and effectively addresses most prior concerns. The analysis is robust, supported by thorough sensitivity checks. However, by addressing the major suggestions mentioned above, including a White British group in PRS evaluation, accounting for ancestry-socioeconomic confounding, and validating findings in external cohorts would further strengthen the study. Minor edits on sample description and tense consistency are also recommended.

(Remarks on code availability)

Reviewer #3

(Remarks to the Author)

(Remarks on code availability)

Reviewer #4

(Remarks to the Author)

All of my concerns have been well addressed.

(Remarks on code availability)

Version 2:

Reviewer comments:

Reviewer #2

(Remarks to the Author)

Joyce Y. Wang et al. outlined three key open questions in polygenic score (PGS) analysis: high variability, trait-specific portability, and measurement impact using UK Biobank data. This version of manuscript demonstrates substantial improvement and successfully resolves most of the earlier concerns. Multiple sensitivity analyses further support the robustness of the study's conclusions.

Overall, my previous concerns have been thoroughly addressed and clearly described in this revision. Congratulations to all co-authors for the nice work.

I look forward to seeing future work inspired by this study, including replication in external cohorts to further validate the findings and assess their generalizability across diverse populations.

(Remarks on code availability)

Reviewer #3

(Remarks to the Author)

(Remarks on code availability)

Wang et al. - Response to Reviews

We thank the four reviewers for their thoughtful assessment and comments. We appreciate the opportunity to improve our work through revision and resubmission.

The reviews clarified two main goalposts for our revision. First, the reviewers had various methodological concerns and questions about the sensitivity of our conclusions to analyses choices. Through a wide range of new analyses, we believe we have addressed these concerns thoroughly. For example, we now show that our conclusions about trait-specificity and measure-specificity extend to the portability of PGS for disease risk prediction. We also show that our main conclusions are robust and insensitive to choices of GWAS covariates and methodology, adjustments for confounding, polygenic scoring methods, measure of genetic distance, and fitting and visualization of continuous functions to describe portability trends.

In addition, one of the reviews conveyed doubts about the significance of our findings. We would like to make the case for why we see our results as substantially advancing the literature. We highlight empirical examples of the drivers and nature of the polygenic score portability problem. This includes crucial conclusions that may be intuitive, but are obscured or severely underappreciated in the current literature. Below, we discuss how our three main conclusions innovate with respect to the current literature:

- (1) Genetic ancestry only explains a negligible proportion of the variation in predictive performance among individuals. Work such as Ding et al., *Nature* 2023 also refers to a “prediction accuracy” at the individual level, and gives the strong impression (e.g., Figures 1d,2c from Ding et al., copied below) that genetic ancestry nearly fully accounts for the variable performance of polygenic scores. (We discuss in our response and in the manuscript (lines 58-66) why we do not consider their statistic as in fact measuring prediction accuracy.)

Taken from figure 1 in Ding et al., *Nature* 2023

Taken from figure 2 in Ding et al., *Nature* 2023

Our paper (lines 16-18, 104-124) shows that for a measure of realized prediction accuracy, the opposite is true: Genetic ancestry explains a small fraction at the individual level (e.g., **Fig. 2B,D,F**); and that SES measures explain a comparable (across traits, slightly larger) fraction (**Fig. 3B**).

Fig. 2: Righthand panels show that genetic distance explains little of the variation in Individual-level prediction accuracy

Fig. 3: SES measures (in particular, Townsend Deprivation Index) and genetic distance explain variance in prediction accuracy comparably well.

- (2) **Trends of portability vary across traits.** While some previous work contain figures where trait-specificity can be observed, we discuss it explicitly for body fat-related polygenic scores and for immune-related polygenic scores. We dove deeper into the latter as a case study, showing how natural selection and ascertainment biases can combine to drive trait-specific trends (Figs. 4, S22-24; lines 133-162; more discussion of implications in lines 199-208).
- (3) **Even qualitative trends of portability can depend on the measure of prediction accuracy used.** To our knowledge, ours is the first work to demonstrate that the choice of measure can entirely change the conclusion about PGS portability and implications for their application. To our previous main text case study of white blood cell count, we have now extended this conclusion to risk prediction, showing that trends look qualitatively different for different measures of type 2 diabetes risk predictive performance—recall and precision (lines 163-182; more discussion of implications in lines 209-219).

We hope that you find our revision and response to reviews to thoroughly address concerns. Thank you for your consideration.

Point-by-Point Response

Reviewers' comments:

Reviewer #1 (Remarks to the Author):

Wang and colleagues use the UK Biobank to highlight what they call three open questions that remain to be understood about the (poor) portability of polygenic scores (PGS). They do not provide answers to any of these three remaining questions and leave this for future work. I am not sure what to get from this paper. I also have concerns about the validity of some of the analyses.

Based on our read of the literature, the three problems are largely missed in the current literature. The impact of factors other than global ancestry on portability, the sensitivity to the measure of portability, and trait specificity are not understood and underappreciated, despite results from Privé et al., *AJHG* 2022, Mostafavi*, Harpak*, *eLife* et al. 2020 and Hou et al., *Nature Genetics* 2024 all being highly cited. This is in contrast to the general impacts of genetic ancestry on portability (LD differences, e.g. Privé et al., Wang et al., *Nature Communications* 2020, Yair and Coop, *Royal Society B* 2023; and allelic turnover, e.g. Carlson et. al., *PLoS Genetics* 2022) which are widely-appreciated and heavily studied. We discuss this concern further in the general introduction to this Response to Reviews, pages 1-4

As for concerns about the validity of analyses, we believe our response below and extensive additional analyses address all outstanding reviewer comments.

Main comments:

- "Previous reports suggested that the relationship between genetic distance and prediction accuracy is similar across traits [17,27,7]". I don't think this is true; these papers present the trend averaged across traits, but also acknowledge differences across traits (e.g. Fig 4 of Ref 7 and Fig 2 of Ref 27).

We agree that this sentence was not sufficiently clear. We meant to say that these references suggested a similar monotonic (in references 7 and 29, also linear) relationship across all traits. We have edited the sentence to clarify. Lines (lines 125-128) now read:

One might expect a qualitatively similar, monotonic relationship between genetic distance and prediction accuracy across traits. Previous analyses (that have not examined individual level prediction accuracy) observed similarly monotonic (Martin et al., 2019), and even linear (Privé et al., 2022, Ding et al., 2023), relationship regardless of the trait examined. However, we observed variation in this relationship among traits. Unlike the case of height, the prediction accuracy for many other traits did not decay monotonically with genetic distance. Weight and body fat percentage peaked in accuracy at intermediate genetic distances.

- At least one of the open questions that the authors mention is why is there some PGS portability issue and why is it different across traits? The consensus on the main reason behind the poor portability of PGS is that we use tagging variants in the PGS (instead of causal variants). These tagging variants then can have different allele frequencies, and especially different LD with the causal variants across ancestries, which can make them bad proxies of causal variants in other ancestries, resulting in a decrease in PGS performance. I don't think this is mentioned in this paper; instead rather vague possible explanations are mentioned, which I think don't help move the field forward.

The previous results on (population genetic) theory-based expectations for portability are discussed explicitly and referenced thoroughly. For example, lines 37-43 states:

This so-called “portability” problem is a subject of intense study. Typically, portability is evaluated through variation in the within-group phenotypic variance explained by a PGS (i.e., the coefficient of determination, R^2) among genetic ancestry groups. Indeed, population genetics theory gives clear predictions for the relationship between genetic dissimilarity to the GWAS sample and PGS prediction accuracy under some models (neutral evolution (Privé et al. 2022, Yair et al. 2022, Carlson et al. 2022), directional (Patel et al. 2024), or stabilizing selection (Yair et al. 2022, Patel et al. 2024), all else being equal (including, e.g., assumptions about environmental effects).

By and large, existing models all predict a monotonic decrease of prediction accuracy with genetic distance. They predict or assume the majority of variation in prediction accuracy will be explained by genetic distance.

We are indeed challenging this so-called “consensus” in this manuscript. While we do not doubt linkage disequilibrium differences between the GWAS sample and the prediction sample are a major factor underlying our and previous observations, we highlight underappreciated factors that add to and interact with ancestry differences. For example: the statistical problems with focusing on R^2 measures, the fact that with some measures prediction accuracy do not even decay with genetic distance, trait-specificity relating to genetic architecture and portability of causal allelic effects.

- I have many concerns on how PGS and genetic distances were derived in this paper.

1/ PGS are derived using the C+T method with $p < 1e-5$. This will likely use a very restricted number of variants, which could result in non-normal PGS distribution, which could cause R^2 to be biased or at least highly variable.

The majority of analyses are about individual-level portability, as we think the view through R^2 can conflate various confounded explanations (lines 44-65).

That said, even in the analysis of R^2 trend there is no assumption that the dependent variable is Normally distributed. We are therefore unclear about the claim above about bias. We are similarly unsure why a large variance of R^2 should take away from our conclusions.

We also note the manuscript is concerned with realized prediction accuracy of PGS as used in practice. It is not concerned with constructing optimal PGS, measuring or improving common practices in polygenic scoring.

Nevertheless, we hope our extensive sensitivity analyses in response to reviewers' comments show that our conclusions are insensitive to many choices in the GWAS (lines 734-752; **Figs. S50-S57**) and construction of PGS (lines 753-765; **Figs. S58-S61**) or analysis (lines 761-783; **Figs. S38-S49; S62-S65**).

Also, I don't understand how the authors can derive individual PGS intervals or accuracies when not using Bayesian sampling of causal effects (as done in Refs 6 & 7).

We measure prediction accuracy with respect to the prediction of trait value. For continuous traits and the individual level, this is the squared difference between the predicted trait value and the true trait value. (We note that in order to not have the analysis affected by how well covariates other than the PGS predict the phenotype, we first regress out covariates; lines 96-103; 349-363).

While Ding et al. also refer to "prediction accuracy," their statistic in fact does not examine the true phenotype value at all, and therefore should not be thought of as the accuracy in prediction of the realized phenotype. Instead, Ding et al.'s statistic is a prediction interval, or its length, i.e. expected uncertainty in prediction under an assumed model. Importantly, it is calculated based on the independent variable (genotype) alone and does not take the dependent variable (phenotype) as input (lines 56-65).

2/ In Ref 27, genetic distance is directly computed from the (squared) Euclidean distance of PCs (which are already weighted by eigenvalues, so I do not think these should be weighted more).

We do not weigh it twice.

3/ Partial- R^2 values between 0 and 4 are odd. Use the true definition of a partial correlation: the correlation between the PGS and the outcome, after they have been both residualized by covariates. You can use e.g. `bigstatsr::pcor()` in R. Why not use the deprivation index in the adjusted covariates?

To make the visualizations of portability trends comparable across traits, we do not show the raw partial R^2 (at the group level) or the raw squared error (at the individual level) on the y-axis (lines 327-363; **Fig. 2** caption). Instead, we show these values divided by “baseline” values for each trait and measure. These baseline values correspond to the average prediction accuracy in prediction sample individuals that are most similar to the GWAS sample individuals, in terms of genetic distance from the GWAS sample centroid. (These baseline values are detailed for continuous traits in **Table S1** and for binary traits in **Table S2**.)

Regretfully, we previously only explained this visualization choice in the Methods section and understand this caused confusion. We now expand on it in the caption of Figure 2 and have edited the figure to call out the fact that relative prediction accuracies are shown.

Figure 2: Trends of portability vary across traits and measures. At the group level (left panels), we measured prediction accuracy with the squared partial correlation between the PGS and the trait value in 500 bins of 258-259 individuals each. At the individual level (right panels), we measured prediction accuracy as the squared difference between the predicted phenotype and the true phenotype value. At both the group and individual levels, y-axis values show relative prediction accuracy, i.e. prediction accuracy divided by a baseline value. The baseline value is the mean prediction accuracy in 50 bins with median genetic distances that are closest to the mean genetic distance for GWAS individuals. **Table S1** details these baseline values for each trait. Curves show cubic spline fits, with 8 knots placed based on the density of data points. **A, B.** For height, prediction accuracy decays nearly monotonically with genetic distance at both the group (**A**) and individual (**B**) levels. **C, D.** For weight, prediction accuracy does not monotonically decay with genetic distance. **E, F.** For white blood cell count, at the group level, prediction accuracy drops near zero at a short genetic distance from the GWAS sample (**E**); yet at the individual level, it increases (**F**). See **Fig. S2-S5** for other traits and **Fig. S6-S9** for plots showing the full ranges of individual-level prediction accuracy.

Also, I don't think it is valid to compute an R^2 in possibly very heterogeneous groups (the same bin on genetic distance can include very different genetic ancestries). If you want to use more groups than in Ref 27, have a look at the many homogeneous non-UK groups defined in <https://doi.org/10.1093/bioinformatics/btac348>.

Our manuscript focuses on individual level prediction. We only use R^2 to discuss the correspondence, and sometimes inconsistency, with assumptions in previous literature. However, computing R^2 is valid regardless of how heterogeneous the group is. We agree that the inference based on the comparison of R^2 between groups can be misleading—with one of the reasons being heterogeneity in genetic variance among groups. This reason is discussed in the introduction, in lines 108-110 of the **Results** section and supported by **Fig. S22**.

44 However, inference based on empirical variation in R^2 can be misleading for various
45 reasons. For one, it can be arbitrarily low even when the model fitted to the data is correct.
46 It also cannot be compared across transformations of the data. R^2 is not comparable across
47 datasets, because, for instance, it depends on the extent of variation in the independent
48 variable^{41,17,36}. In the context of inference about the causes of PGS portability, these issues
49 can manifest in different ways. For example, heterogeneity in within-group genetic variance
50 and environmental variance can each greatly affect group differences in R^2 .

Figure S22: Mean heterozygosity of SNPs, stratified by effect size. This figure presents the same analysis as Fig. 4B in the main text, but for showing data for the 12 quantitative traits not included there. For each trait, SNPs are stratified into three equal-sized strata (small, medium, and large) based on squared effect sizes (Fig. S23). Each data point is the mean heterozygosity of a stratum in a bin of genetic distance.

4/ Use only 16 PCs from the UK Biobank; PC19-40 capture only LD and should never be used (cf. <https://doi.org/10.1093/bioinformatics/btaa520>).

To address this concern, we recalculated of genetic distance, now defining it using the first 16 rather than 40 PCs. None of the portability trends and our conclusions based on them changed (lines 712-717; lines 761-773; Figs. S62-S65). As an example, we show below a comparison of both individual- and group-level results for red blood cell-related traits with the two measures of genetic distance.

5/ I don't think there has been any filtering on relatedness in these analyses, which can bias predictions for related individuals.

Thanks to reviewers' attention on this topic, we discovered a mistake in our code resulting in relatives not being properly excluded from analysis. We fixed the mistake and close relatives are removed throughout now. In particular, we filtered by conditioning on individuals that were used in the PCA calculation of the UKB (i.e., relying on the original filtering of individuals with $\leq 3^{\text{rd}}$ degree relatedness, performed by the UKB; lines 241-242). As a result, the composition of the GWAS sample and prediction sample and downstream analyses (effectively all analyses) have been updated. Importantly, our conclusions remain the same.

Additional comments:

- Not sure it is valid to compute F_{st} between one individual and a group.

It is valid. F_{st} is defined with respect to two subsets of chromosomes (from a larger set). Here, one of the subsets is composed of only two chromosomes.

- Discussion materials should be more separated from the Results.

In a paper such as this one, aiming to highlight underappreciated facets of a problem, we view signposting, including minimal interpretation of results that comes alongside their reporting, as helpful to understanding and evaluating our findings.

- "We standardized genetic distance such that its mean is 1 across GWAS sample individuals." -> what is the rationale behind this? Instead, you could scale using the max distance and max F_{st} , as done in Ref 27.

Signed Florian Privé

We apologize that this was unclear, and have edited the text to better explain this choice (**Fig. 2** caption; lines 266-310). In short, this is a division of genetic distance by a constant. The rationale is for the genetic distance to have interpretable units. Our scaling is such that a genetic distance of 3 means three times further than a “typical GWAS sample individual” from the GWAS centroid. To do that, the constant we divide the raw genetic distance by is simply the mean (raw) genetic distance among GWAS sample individuals. In Fig. 2C, for intuition, we also show the values of project 1000 genomes populations on this scale.

Figure 1: Measuring “genetic distance” from the GWAS sample. **A.** Across 336,923 individuals in the GWAS sample and 69,500 individuals in the prediction set, we measure “genetic distance” from the GWAS sample as the weighted Euclidean distance from the centroid of GWAS individuals in PCA space, with each PC weight being proportional to its respective eigenvalue. **B.** Across 10,000 individuals from the prediction set, genetic distance to the GWAS sample (calculated with 40 PCs) is highly correlated with F_{st} between the GWAS sample and the individual (**Fig. S1**). Under a theoretical model where portability is driven by genetic ancestry alone and the trait evolves neutrally, F_{st} should perfectly predict variation in prediction accuracy. We note that genetic distance is less reflective of F_{st} for intermediate genetic distances. **C.** The distribution of genetic distance. For reference, we show the mean genetic distances for subsets of the 1000 Genomes dataset³: CEU, Utah residents of primarily Northern and Western European descent; CHB, Han Chinese in Beijing, China; YRI, Yoruba in Ibadan, Nigeria. The dashed line represents the 97.5th percentile of genetic distance from among GWAS sample individuals. In what follows, our reports are based on individuals with genetic distances larger than this value. The inset is a zoomed-in view of a smaller range and on a log-scale, to better visualize the distribution within the prediction sample.

Reviewer #2 (Remarks to the Author):

Review for “Three Open Questions in Polygenic Score Portability”

Joyce Y. Wang et al described three open questions for PGS analysis, including high variability, trait-specific portability and measurement impact. Overall, the study highlights that understanding PGS portability requires considering more than just global ancestry, including trait evolution, genetic architecture, social

factors, and PGS construction. This approach could lead to more effective and equitable genomic research.

The study topics are very interesting. Manuscripts are well-written with clear illustration. Detailed major comments can be found in the following section.

We thank the reviewer for their thoughtful evaluation and comments.

Major:

1. A “table 1” should be included to describing the distribution of data set. Since authors examined variables like Townsend Deprivation Index, Household Income etc., it would be necessary to compare between GWAS sample and PGS sample to make sure there is no significant differences between the group. Besides variables used in the analysis, some other variables should be included in table 1 include age, sex, location (England, Scotland or Wales).

We have added detailed analyses of the distributions of age, sex and location in the GWAS and prediction samples (lines 695-711; **Figs. S25-S29**).

The fact that the distribution of age, sex, location and other environmental and social factors may differ between the GWAS sample and the prediction sample is indeed what we want to highlight here. As shown in e.g. Mostafavi et al., *eLife* 2020 and Hou et al., *Nature Genetics* 2024, age, sex and SES differences can greatly impact prediction accuracy even in the absence of major genetic ancestry differences. Here, we wish to highlight that the two groups of factors are also often confounded.

2. Line 99-100 & Section Prediction accuracy is weakly predicted by genetic distance & Figure 2: Authors split samples into a bin of approximately 260 individuals and compared the prediction accuracy against the genetic distance from the GWAS sample. (1) In Figure 2B regarding weight, the smoothed curve around 100 on x-axis seems to be influenced by a few points on the top. Some further analysis could be done to rule out the randomness of the sample split. For example, would that be possible to vary the number of bins (currently 500) to see if the current trend would maintain.

To examine the sensitivity of our analyses to binning parameter choices, we re-performed the analysis of portability trends with a different partitioning: we doubled the number of (half-size) bins. Namely, we binned the individuals in the prediction set into 500 bins of 139 individuals each, instead of 250 bins of 278 individuals each (the partitioning which is now used in the main text analyses). In **Figs. S38-S41**, we show relative prediction accuracy trends using the mean prediction accuracy in 50 bins with median genetic distances that are closest to the mean genetic distance for GWAS individuals as the baseline value. Comparing the two binning choices, we did not notice any substantial change in the portability trends for any of the quantitative traits analyzed (including weight—see panels C,D in **Fig. S38** and attached below; lines 721-728).

(2) Additionally, currently analysis uses cubic spline fits with 8 knots to capture the trajectory of the trend. It would be helpful to fit some models, like GAM (generalized additive model) to show if the trend is significant or not. And adding the CI (confidence interval) around the smoothed line.

To examine whether our conclusions are sensitive to continuous function fits we applied for visualizations, we fitted a generalized additive model (GAM) instead of a cubic spline. The GAM fits appear to be smoother at the group level for most traits, but have the same general trends as the cubic spline fit (Figs. S42-S45; lines 779-783).

To address the concern about the number of knots used in the spline, we have also reperformed the analyses using cubic splines with 12 instead of 8 knots. Here too, we found no qualitative changed to the trends we observed previously and discuss in the text (Figs. S38-S41; lines 774-778).

(3) Figure 2 500 bins plot with individual ones are not quite corresponding to each other. The overall trend between the 2 are not consistent. For example, E and F, the whole trajectory in E is below 1. However, in F the line is above 1.

Indeed, one of our main findings is that portability trends depend on the measure of prediction accuracy. In Fig. 2E,F, this is exemplified with a measure of group-level prediction accuracy compared with a measure of individual-level prediction accuracy. See specifically the section titled “The measure of predictive performance can alter our view of portability” in lines 163-167. In the new Figure 5b, we can see the same can happen for different measures of type 2 diabetes risk prediction accuracy, recall and precision.

3. Line 183-189 in Discussions: Authors observed some trait-specific trends in portability. It would be great to have some additional analysis to examine what groups of traits might need additional inspections. Expanding on the potential implications of these findings for equitable genomic research could strengthen the conclusion.

We have exemplified trait-specificity for various traits, including body fat related traits (lines 129-132), immune related traits (lines 133-162) and type 2 diabetes (168-182).

The unusual patterns observed for immune-related traits include the rapid decrease in prediction accuracy even at short genetic distances. We propose a testable hypothesis:

1. When selection pressures vary across time/geography, such as would be expected for the immune system, there will be a more rapid turnover of the loci contributing to trait variation. Genetic effects themselves become less portable. Indeed, when comparing to other traits (including a trait with similar SNP heritability), we find evidence that genetic effects on lymphocyte count are less portable.
2. The turnover in genetic basis across populations can interact with statistical biases, resulting in another unusual pattern: the frequencies of the largest effect index SNPs are increasing with genetic distance.
3. Since these alleles make for a predictor that is highly variable away from the GWAS sample (due to (2)) but quickly becomes less predictive (due to (1)), prediction accuracy decays quickly.

Following the reviewer's suggestion, we expanded the discussion of potential implications in the Discussion section (lines 199-208). We do not want to frame our conclusions as prescriptive in terms of increasing equity in genomic research. However, we strongly agree that understanding the importance of social, environmental and evolutionary context for PGS portability is crucial for equitable genomic research. This is a central motivation for this work (lines 11-13; 28-29; 34-36; 218-219).

4. Line 287-297: Authors detailed described how GWAS are constructed within UKB. (1) Did authors also take sample relatedness into consideration? Either model the sample relatedness or excluded those individuals that are closely related. These individuals might influence GWAS performance. Some more recent software, like REGENIE would be a better approach for modeling GWAS.

Thanks to reviewers 1 & 2 comment and attention here, we discovered that, due to an error in our code, relatives were not properly removed. We fixed the bug and relatives are now removed. In particular, we kept just one of each set of individuals with $\leq 3^{\text{rd}}$ degree relatedness (lines 241-242). As a result, the composition of the GWAS sample and prediction sample and downstream analyses (effectively all analyses) have been updated. Nevertheless, in the vast majority of analyses, our conclusions remain the same.

In addition, to test the sensitivity of our conclusions to the GWAS method we used, in particular the adjustment for population structure using PCs, we reanalyzed portability trends for PGS constructed

based on GWAS performed with REGENIE. REGENIE adjusts for population structure using a linear mixed model approach. Other research from us (Smith et al. 2025, “A litmus test for confounding in polygenic scores”) and others suggests that this type of adjustment better mitigates some types of confounding, such as confounding due to finer-scale population structure.

Prediction accuracies were generally higher throughout in REGENIE GWAS-based polygenic scores. In a couple of the lower heritability traits (lymphocyte count and monocyte count), portability trends became noisier. Beyond these differences, trends remained the same (lines 740-747; **Figs. S54-S57**).

Figure S55: Trends of portability with GWAS run using regenie, for white blood-cell related traits. This figure presents the same analysis as **Fig. 2** and **Fig. S2**, but the first and third columns (**A, C, E, G, I, K, M, O**) are plots with GWAS run with regenie. The second and fourth columns (**B, D, F, H, J, L, N, P**) contain plots with GWAS run with PLINK, taken from the main text or supplementary text for comparison. At the group level (left two columns), we measured prediction accuracy with the squared partial correlation between the PGS and the trait value in 250 bins of 278 individuals each. At the individual level (right two columns), we measured prediction accuracy as the squared difference between the predicted phenotype and the true phenotype value. At both the group and individual levels, y-axis values show relative prediction accuracy, i.e. prediction accuracy divided by a baseline value. The baseline value is the mean prediction accuracy in 25 bins with median genetic distances that are closest to the mean genetic distance for GWAS individuals. **Table S1** details these baseline values for each trait. Curves show cubic spline fits, with 8 knots placed based on the density of data points.

(2) UKB data is collected at 3 regions, ie England, Scotland and Wales, in multiple centers in different batch. It would be great to also control assessment center (field id 54) and genotype array (field 22000) in the analysis.

We explored the distributions of these different sample characteristics and their variation with genetic distance in lines 695-711 and **Figs. S25-S29** (**Fig. S25** in particular shows country of residence and is copied below).

Per the reviewer's concern, we re-performed the analysis of portability trends across the 15 quantitative traits using genotype array and participant country of residence as additional covariates in the GWAS. We find that this results in lower PGS prediction accuracy for lymphocyte and monocyte counts, but makes no difference to the qualitative portability trends reported throughout (lines 729-739; **Figs. S50-S53**).

5. Line 298-300: Authors use clumping and thresholding method to construct PGS model. Would that be possible to utilize other methods, like LDpred2 and PRSCS to examine the performance of PGS? Literatures have showed that these methods show a better performance comparing to clumping and thresholding strategy.

We performed an analysis to test sensitivity to the choice to analyze PGS constructed using clumping and thresholding by replacing it with PRS-CS (Ge et al., *Nature Communications* 2019). PRS-CS applies Bayesian shrinkage to GWAS-based allelic effect estimates. It also aims to consider LD patterns more rigorously rather than via the clumping and thresholding heuristic.

We compared our results for the 15 clumping and thresholding based polygenic scores for quantitative traits analyzed in the main text. For weight and BMI at the group level, the peak at intermediate genetic distances became less prominent for polygenic scores constructed with PRS-CS. For lymphocyte count and white blood cell count, the decrease in prediction at the group level became more gradual. However, the general trends of portability of all other traits did not change (lines 748-760; **Figs. S58-S61**).

Minor:

1. Figure 1C: would that be possible to have a similar histogram for UKB samples?

The samples used to produce the histograms in Fig. 1C are UKB samples.

Figure 1: Measuring “genetic distance” from the GWAS sample. **A.** Across 336,923 individuals in the GWAS sample and 69,500 individuals in the prediction set, we measure “genetic distance” from the GWAS sample as the weighted Euclidean distance from the centroid of GWAS individuals in PCA space, with each PC weight being proportional to its respective eigenvalue. **B.** Across 10,000 individuals from the prediction set, genetic distance to the GWAS sample (calculated with 40 PCs) is highly correlated with F_{st} between the GWAS sample and the individual (**Fig. S1**). Under a theoretical model where portability is driven by genetic ancestry alone and the trait evolves neutrally, F_{st} should perfectly predict variation in prediction accuracy. We note that genetic distance is less reflective of F_{st} for intermediate genetic distances. **C.** The distribution of genetic distance. For reference, we show the mean genetic distances for subsets of the 1000 Genomes dataset³: CEU, Utah residents of primarily Northern and Western European descent; CHB, Han Chinese in Beijing, China; YRI, Yoruba in Ibadan, Nigeria. The dashed line represents the 97.5th percentile of genetic distance from among GWAS sample individuals. In what follows, our reports are based on individuals with genetic distances larger than this value. The inset is a zoomed-in view of a smaller range and on a log-scale, to better visualize the distribution within the prediction sample.

2. It would be interesting to examine relationship between genetic PCs and F_{st} estimated in the study.

The relationship between F_{st} and genetic distance is shown in **Fig. 1B**. We added a supplementary analysis of the relationship between F_{st} and individual genetic PCs (lines 712-717; **Figs. S30-S33**)

3. Line 171-176 & Discussions: It would be great to examine some admixed ancestry population performance considering the mosaic genetic structure. F_{st} is still a measure of genetic distance in a global level.

We agree and are similarly interested in examining whether measures of local genetic ancestry can help explain portability trends. However, we view the hefty analysis required to ask this question as beyond the already substantial scope of this manuscript. We discuss this as a fruitful area for future research in lines 187-195.

4. Line 340: field should be 22189 instead of 189.

Fields 22189 and 189 are both Townsend index. 189 is what we used it. Examining the UKB showcase, both data fields have the same distribution. The only difference appears to be that 189 has more decimal places.

In all, the manuscript described important questions in PRS analysis that GWAS summary statistics have different characteristics, especially ancestry with individual level data. While the manuscript is well written and ideas are new, some detailed information need to be refined.

We thank the reviewer for their helpful comments and kind words about the manuscript.

Reviewer #3 (Remarks to the Author):

Reviewer #3 (Remarks on code availability):

Scripts are well-written and easy to replicate.

We thank the reviewer for their kind note regarding replicability.

Reviewer #4 (Remarks to the Author):

Wang et al. highlight three gaps for PGS portability with comprehensive analysis in the UKB. But I still have some concerns.

1. You only used 15 highly heritable continuous traits to study PGS's portability. To my knowledge, the selected traits are polygenic. I assume that the genetic architecture may be a contributing factor to PGS portability. I would like you to add the binary traits, such as type II diabetes (T2D) and Alzheimer's disease (AD). For example, the genetic architecture of AD might be a mixture of polygenic and sparse. Both the difference between continuous trait and binary trait and the difference between genetic architecture might result in different trends between PRS and genetic distance.

There are some clear theoretical expectations for continuous traits under simplified models. However, to our knowledge, no clear predictions exist for binary disease traits. At the same time, we agree it is of interest to examine whether our conclusions hold for disease traits.

In particular, the conclusions about trait-specific, measure-specific portability trends are important to test for binary disease traits, because the choice of predictive performance measures in disease traits can link directly with an intervention or policy that is being considered. For example, when a broadly accessible, safe intervention exists, we might prioritize the identification of all cases over all else and therefore focus on recall (number of true positives divided by number of true cases). In other cases, for example when the intervention includes a risky treatment, we might prioritize avoiding false positives and hence focus on precision (true positives over all positive classifications). Therefore, we sought to know whether trends of portability depend on the measures of predictive performance in disease.

For asthma, we saw that precision and accuracy depended on genetic distance in a qualitatively similar way. For type 2 diabetes, precision appeared roughly constant for medium and large genetic distances, while recall generally increased with distance far from the GWAS sample (lines 168-182; Fig. 5). Therefore, predictive performance shows trait-specific, measure-specific trends in disease risk prediction as well.

We note that we chose these diseases since they are among the very few that had a sufficient number of cases for the analysis to be interpretable. For example, for Alzheimer Disease, which was requested by

the reviewer, we had to increase the size of each genetic ancestry bins substantially such that each bin would contain positive cases; yet that led to a small number of bins overall—each capturing a wide range of genetic distances—making the trends across bins uninterpretable.

2. You only used a simple method to construct PGS, which might result in a low prediction performance. I recommend incorporating a PGS method into the sensitivity analysis. Based on your pipeline, you can use polygenic methods without a validation set, such as SBLUP and DSBLMM. It's intriguing to see if a better PGS yields a different trend for the same trait.

Overall, our manuscript is not concerned with an optimal construction of polygenic scores. It discusses general trends of portability of (imperfect) polygenic scores. They are built in a manner largely reflective of common practice today, albeit not necessarily some gold standard. The explanation for the variable portability trends may very well partly relate to their suboptimal construction.

That said, it is important to show, as the reviewer alludes to here and elsewhere, that our qualitative conclusions are insensitive to specific choices we have made during the analysis.

Per the reviewer's suggestion, we re-analyzed portability trends after constructing PGS with another method that is in broad use in human genetics, PRS-CS (Ge et al., *Nature Communications* 2019; lines 748-760). PRS-CS applies Bayesian shrinkage to GWAS-based allelic effect estimates. It also aims to consider LD patterns more rigorously and holistically, rather than via the clumping and thresholding heuristic.

We first focused on a comparison with the results for the 15 clumping and thresholding based polygenic scores for quantitative traits analyzed in the main text. For weight and BMI at the group level, the peak at intermediate genetic distances became less prominent for polygenic scores constructed with PRS-CS. For lymphocyte count and white blood cell count, the decrease in prediction at the group level became more gradual. However, the general trends of portability of all other traits did not change (Figs. S58-S61).

3. Based on the partial R2 equation, I assume that the value should be lower than 1. I'm confused as to why some groups' partial R2 of weight, BMI, and body fat percentage is greater than 1. I also suggest that you provide the raw R2 to demonstrate PGS's performance.

To make the visualizations of portability trends comparable across traits, we do not show the raw partial R2 (at the group level) or the raw squared error (at the individual level) on the y-axis. Instead, we show these values divided by “baseline” values for each trait and measure. These baseline values (which are shown in **Tables 1-2**) correspond to the average prediction accuracy in prediction sample individuals that are most similar to the GWAS sample individuals, in terms of genetic distance from the GWAS sample centroid.

We apologize that this visualization choice was previously only explained in the Methods section. We understand this caused confusion. We now spell this out in the caption of Figure 2 and have edited the figure to call out the fact that relative prediction accuracies are shown.

Phenotype	SNP h^2	Mean squared prediction error (MSE)	Variance of residualized phenotype ($Var[Z]$)	$1 - \frac{MSE}{Var[Z]}$	Prediction accuracy (partial R^2)
Height	0.4852	31.8859	41.5853	0.2332	0.2473
Cystatin C level	0.3214	0.0319	0.0333	0.0426	0.0632
Platelet count	0.3079	2950.5930	3369.6670	0.1244	0.1365
Mean corpuscular volume	0.2667	16.5441	18.7806	0.1191	0.1303
Weight	0.2654	205.3049	217.1554	0.0546	0.0620
Mean corpuscular hemoglobin	0.2530	2.6206	2.9492	0.1114	0.1253
BMI	0.2482	23.7059	24.6588	0.0386	0.0442
Red blood cell count	0.2337	0.1061	0.1161	0.0859	0.0912
Body fat percentage	0.0472	41.7102	43.2307	0.0352	0.0419
Monocyte count	0.2305	0.03674	0.0380	0.0325	0.0870
Triglyceride level	0.2182	0.9468	1.0010	0.0542	0.0620
Lymphocyte count	0.2103	1.1740	1.1753	0.0011	0.0342
White blood cell count	0.1910	3.8944	4.0020	0.0269	0.0455
Eosinophil count	0.1840	0.0195	0.0204	0.0430	0.0505
LDL cholesterol level	0.0825	0.6270	0.6810	0.0792	0.0887

Table S1: Characteristics of continuous traits and PGS analyzed. SNP heritabilities (SNP h^2) are taken from the Neale Lab’s UKB analysis²⁶. For the 25 bins with a genetic distance most similar to the mean genetic distance of the GWAS group, we calculated the group-level prediction accuracy (partial genetic correlation of the PGS and the trait value), mean individual prediction error (squared prediction error), the variance of residualized phenotype, and the ratio between the two subtracted from one, as another measure of phenotype variance explained by the PGS close to the GWAS sample.

Figure 2: Trends of portability vary across traits and measures. At the group level (left panels), we measured prediction accuracy with the squared partial correlation between the PGS and the trait value in 250 bins of 278 individuals each. At the individual level (right panels), we measured prediction accuracy as the squared difference between the predicted phenotype and the true phenotype value. At both the group and individual levels, y-axis values show relative prediction accuracy, i.e. prediction accuracy divided by a baseline value. The baseline value is the mean prediction accuracy in 25 bins with median genetic distances that are closest to the mean genetic distance for GWAS individuals. **Table S1** details these baseline values for each trait. Curves show cubic spline fits, with 8 knots placed based on the density of data points. **A, B.** For height, prediction accuracy decays nearly monotonically with genetic distance at both the group (A) and individual (B) levels. **C, D.** For weight, prediction accuracy does not monotonically decay with genetic distance. **E, F.** For white blood cell count, at the group level, prediction accuracy drops near zero at a short genetic distance from the GWAS sample (E); yet at the individual level, it increases (F). See **Figs. S2-S5** for other traits and **Figs. S6-S9** for plots showing the full ranges of individual-level prediction accuracy.

4. The variance of PGS for WBC, lymphocyte count, and monocyte count increases with genetic distance. Please conduct further analysis to delve deeper into the trend, particularly examining the specific genetic architecture of immunity-related traits. If you add asthma, an immunity-related trait, the trend should be similar.

We discuss the genetic architecture (joint distribution of allelic effects and allele frequencies), the differences in heterozygosity trends across different effect sizes and hypothesize how the increase with PGS variance may be driven by variable selection pressures on the immune system (lines 133-162; 199-208; **Figs. 4, S22-S24, S66**).

Per the reviewer's suggestion, we have examined the case for asthma. We found arguments in the literature supporting the hypothesis of substantial ancestry specificity of genetic associations with asthma (e.g., Washington III et al., *Genome Medicine* 2022). However, for asthma, this does not appear to be a strong enough effect to reverse the relationship between the variance of asthma PGS and genetic distance. Namely, it does not follow suit with that observed for white blood cell count or lymphocyte

count. Instead, like the vast majority of quantitative traits and like the other disease we analyzed, type 2 diabetes, we see a monotonic decrease in genetic variance with genetic distance (Fig. S66).

5. You investigate the heterozygosity and SNP effect size in different traits. I assume that the LD structure is also a factor in PGS portability. Please perform additional analysis of LD structure, such as LD score.

We apologize, but we were unsure what analysis the reviewer is asking to see and what question it would address.

To address the more general question, yes, absolutely: LD differences between the GWAS sample and prediction sample are plausibly among the most important factors leading to the decay of prediction accuracy with genetic distance. We discuss this fact throughout, including the previous theoretical and empirical support and their caveates (lines 37-43; 51-55) and consistency of this prediction with our observation (lines 104-115; 515-159; 187-195). In this manuscript, however, we show that summaries of genetic ancestry alone (including allele frequency turnover and LD differences) explains little of the variation in predictive performance at the individual level (Fig. 2B,D,F; Fig. S2-S9; Fig. 3B).

Wang et al. - Response to 2nd Round Reviews

We thank all reviewers for their valuable comments and assessment of our work. We have revised our work further to address outstanding comments. Please find below a point-by-point response which also details our additional revisions.

REVIEWER COMMENTS

Reviewer #1 (Remarks to the Author):

I am still unconvinced by this paper and have several comments.

We regret not being able to meet the expectations of the reviewer. At the same time, we note that, as detailed in the previous Response to Reviews—and not contested by any of the reviewers—we have addressed each and every one of the concerns and requests for additional analysis raised in the previous round of review. Numerous sensitivity analyses performed in the previous round of revision, and additional ones carried out in this round, support the robustness of our observations and conclusions.

Main comments

- Gap 1 & 3: Results (trends of predictive performance vs genetic distance to training) using the prediction error at the individual level do not agree with the ones at the group level. This is very confusing (I guess this is the authors' point). The problem I see is that the squared difference in predicted vs true value depends heavily on things like phenotypic variance and also random shifts in PGS distribution (which are corrected for when using some proper partial correlation at the group level).

This are indeed central conclusions of our work, including both the improperness of R^2 for most applications and its limited interpretability (lines 44-55; 164-168; 210-211); and the dependence on the measure of predictive performance, discussed in the Results subsection entitled “The measure of predictive performance can alter our view of portability.”, lines 164-183, and in the Discussion, lines 210-220. To our knowledge, ours is the first work to demonstrate that the choice of measure can entirely change the conclusion about PGS portability and implications for their application. In our previous revision, we extended this conclusion to risk prediction, showing that trends look qualitatively different for different measures of type 2 diabetes risk predictive performance—recall and precision (lines 164-183; Fig. 5).

- To get better quality results, the authors should present results
+ with PRS-CS instead of C+T everywhere given the higher number of variants used with make the results less noisy (C+T could simply pick a few highly-transferable genetic variants by chance)

We performed an analysis to test the sensitivity of our conclusions to the choice to analyze PGS constructed using clumping and thresholding and replacing it with PRS-CS (Ge et al., Nature

Communications 2019). PRS-CS applies Bayesian shrinkage to GWAS-based allelic effect estimates. It also aims to consider LD patterns more rigorously rather than via the clumping and thresholding heuristic.

We find that, for weight and BMI, the peak observed in the group level analysis at intermediate genetic distances became less prominent for polygenic scores constructed with PRS-CS. For lymphocyte count and white blood cell count, the decrease in prediction at the group level became more gradual. However, the trends of portability of all other traits, and our qualitative conclusions reported in the main text, did not change (Figs. S58-S61).

In the examples (from Fig. S58) shown below, odd columns show the new sensitivity analysis with PRS-CS polygenic scoring; even column show main text choice of the GWAS and prediction samples.

Figure S58: Trends of portability with PGS constructed using *PRS-CS*, for anthropometric measurements. This figure presents the same analysis as Fig. 2 and Fig. S2, but the first and third columns (A, C, E, G, I, K, M, O) are plots with PRS reconstructed with *PRS-CS*. The second and fourth columns (B, D, F, H, J, L, N, P) contain plots with PGS constructed with *PLINK*, taken from the main text or supplementary text for comparison. At the group level (left two columns), we measured prediction accuracy with the squared partial correlation between the PGS and the trait value in 250 bins of 278 individuals each. At the individual level (right two columns), we measured prediction accuracy as the squared difference between the predicted phenotype and the true phenotype value. At both the group and individual levels, y-axis values show relative prediction accuracy, i.e. prediction accuracy divided by a baseline value. The baseline value is the mean prediction accuracy in 25 bins with median genetic distances that are closest to the mean genetic distance for GWAS individuals. Table S1 details these baseline values for each trait. Curves show cubic spline fits, with 8 knots placed based on the density of data points.

+ with GAM instead of the cubic splines, which is much more robust and informative; you can also consider log-distances to have more evenly spaced bins

To examine whether our conclusions are sensitive to continuous function fits we applied for visualizations, we fitted a generalized additive model (GAM) instead of a cubic spline fits. Trends remained qualitatively the same as with the cubic spline fit (Figs. S42-S45).

+ the authors should really consider log-transforming all the phenotypes they used (before residualizing) as they are probably all log-normally distributed (even height to a lesser extent -> just have a look at the distributions of males and females); this would make their result more robust to the long right tails of the phenotypic distributions.

We appreciate the suggestion and view the question as an interesting theoretical investigation. However, it almost opposite to our empirical, applicative focus in this work. The sensitivity of group level performance metrics to the population and prediction sample distributions is the point, not an artifact.

More broadly, our work is motivated and concerned with applicative performance, as opposed to optimizing the statistical capturing of variance in a theoretical construct that is more mathematically convenient. In central applications of polygenic scores, one typically aims to predict either the quantitative phenotype value in its given, defined units; or a categorical disease state. Prediction of some transformed phenotype, where the transformation depends on some reference sample, means missing this aim.

+ consider whether the individual-level predictions are just simply shifted due to differences in allele frequencies, which is corrected when adjusting PGS for PCs (cf. first main comment).

We agree with this intuition, and this is indeed part of the rationale why we are indeed adjusting for genetic distance in our individual level. Genetic distance correlates strongly with F_{st} (Fig. S1) and with top PCs, predominantly with PC1 (Fig. S30-S33)

- Gap 2: “portability trends (with respect to genetic distance) can be trait-specific” -> again, this has been shown in previous papers and I don't consider this as a knowledge gap. Moreover, I think the use of C+T with just a few variants in the PGS make them even more trait-specific simply by chance of picking +/-

transferable genetic variants, which I consider misleading (therefore the recommendation of using a more genome-wide PGS method like PRS-CS).

This comment is addressed extensively in the previous Response to Reviews, revision and in the comment above.

Additional comments:

- The figures in the response to reviewers were not of particularly good quality.
- Ding et al. have looked at individual-level prediction accuracy; that was the whole point of the paper. They used some estimates, but still.

We are unsure what is meant by “they used some estimates, but still,” but we would like to clarify this is not a matter of more or less precise estimation of an estimand.

In this work, we measure prediction accuracy with respect to the prediction of trait value. For continuous traits and the individual level, this is the squared difference between the predicted trait value and the true trait value. While Ding et al. also use the same terminology of “prediction accuracy,” their “prediction accuracy” statistic in fact does not examine the true phenotype value at all. Indeed, it is not the accuracy in prediction of the realized phenotype. Instead, Ding et al.’s statistic is a prediction interval, or its length, i.e. expected uncertainty in prediction under an assumed model. Importantly, it is calculated based on the independent variable (genotype) alone and does not take the dependent variable (phenotype) as input (lines 56-65).

We note that our understanding of this different use of has been confirmed with the first and last (corresponding) authors Ding et al. through personal correspondence.

- “The previous results on (population genetic) theory-based expectations for portability are discussed explicitly and referenced thoroughly.” -> which still says nothing on LD and AF differences..

With apologies, we do not understand this comment/question/concern. We would need for it to be clarified further in order to address it.

Reviewer #2 (Remarks to the Author):

Joyce Y. Wang et al described three open questions for PGS analysis, including high variability, trait-specific portability and measurement impact. The manuscript had a great improvement since the last version. Majority of previous issues that were raised last time are properly addressed. Various sensitivity analyses validated the robustness of analysis conclusions. Here are some additional major and minor suggestions for authors to consider:

Some major changes could be:

1. Line 76, fit of GWAS and PRS analysis: authors subsetted 336923 individuals for GWAS analysis and applied to prediction sample for PRS. Notably, the GWAS was conducted exclusively in White British individuals, while the PRS was evaluated in a non-White British population. Given the known impact of population structure and genetic ancestry on PRS transferability, this discrepancy raises concerns about the validity of cross-ancestry comparisons. It would strengthen the analysis and interpretation if the authors could also include a subset of White British individuals in the PRS prediction phase. Comparing PRS performance between White British and non-White British individuals—using the same GWAS-derived weights—would provide a more rigorous assessment of ancestry-related biases and enhance the robustness of the conclusions. In that case, including the White-to-White group as a baseline would serve as a fundamental point of comparison.

To address the reviewer's concern, we performed an additional sensitivity analysis, in which we considered a different choice of a GWAS sample and prediction sample, moving a subset of the set labeled by the UKB as White British into the prediction sample. In particular, the GWAS sample in this analysis included 300,000 randomly selected White British individuals and the remaining 36,923 White British individuals were included in the prediction sample (Text S3, lines 253-269). We find that among these individuals, whose genetic distance from the GWAS centroid is typically lower than 3 (Fig. S75), we find that group-level prediction accuracy remains approximately constant, as we have seen in the main text analyses (Figs. S76-S79). We note that the White British subset indeed spans a very narrow range of genetic distance (Fig. S71 and below).

As a result, we also find that qualitative trends of portability as a function of (the full range of) genetic distance remain qualitatively unchanged for all traits: See Figs S72-S75 and examples pasted below (odd columns show the new sensitivity analysis with a prediction sample that includes White British individuals; even column show main text choice of the GWAS and prediction samples).

2. Currently, all analyses—including both GWAS and PRS—are conducted within the UK Biobank. To strengthen the generalizability of the findings, it would be valuable to assess external validity using independent cohorts. For example, applying GWAS results from an external dataset to train PRS and testing them in the UK Biobank, or vice versa, could help rule out potential UK Biobank-specific cohort

effects and enhance the robustness of the conclusions. Some potential large biobank can be used include: All of Us etc.

We do not expect our results generalize in the sense that we will see the same trends of portability with ancestry across biobanks. This is precisely because of the sensitivities the reviewer alludes to. Namely, our main conclusions are about the low fraction of variation in realized prediction accuracy explained by ancestry (which will lead to weak replication of trends across studies in itself), sensitivity to social and environmental factors (which will differ greatly across countries and studies with different recruitment strategies), trait-specificity and measure-specificity (both varying across studies as well).

We sought to illustrate these sensitivities further and address the question raised by the reviewer—albeit through a different analysis than the one suggested by them. In particular, we expanded our analysis comparing the explanatory power of genetic distance to that of measures of socio-economic status. In the new analysis, we stratified our prediction sample by Townsend Deprivation Index, and investigated portability as a function of genetic distance within each stratum independently (Text S3, lines 241-252). Our results further illustrate that, across traits, the explanatory power of each of the factors (SES vs. genetic ancestry) is comparable (Fig. 3B) and that PGS prediction accuracy tends to be higher for more deprived individuals (Fig. 3A,C,S67-70).

We enthusiastically share the interest in learning about variation in portability through empirical, including cross-biobank comparisons of portability. Our lab is therefore investing considerable effort and funds in gaining access to multiple biobanks, and carrying out these analyses in datasets such as All of Us. However, primarily for the reason discussed above, we view these analyses as beyond the scope of this

work—which already addresses many novel questions and supported by a diverse array of sensitivity analyses.

Minor issues:

1. Line 76-78: please explain how the GWAS sample and prediction sample divided.

This is discussed in further depth in the Data Overview subsection of the Methods, lines 233-250. We include the most relevant part below.

243 closer (data field 22020). In total, we removed 96,067 individuals. In the selection of the
244 GWAS sample, individuals who passed all filtering steps and were labeled by the UKB
245 as “White British” (WB)—those who self-identified as “White” and “British” and closely
246 clustered together in PC space (336,923 individuals, data field 22006)—were included in the
247 GWAS sample. The remaining 69,500 individuals who passed filtering (Non-White British,
248 NWB) were used as the prediction set. In the **Supplementary Materials**, we investigate
249 the sensitivity of our analysis to a different partitioning between the GWAS sample and
250 prediction set.

2. Please keep tense consistency through the manuscript. Some places present tense (e.g. line 114) while some locations use past tense (e.g. line 113).

Thank you for pointing out this oversight. We have reviewed the manuscript and supplement and believe we have caught all previous misses (in particular, in the main text, this includes lines 75,98,114). Overall, we used past tense when describing our work and observations. We only use present tense when describing general truths. This follows from several style guides, including Council of Science Editors (“Describe your methods and results in the past tense, since they occurred. Use the present tense for conclusions and generally accepted facts”) and Nature Portfolio Author Guide “Use past tense for your experiments and results..., and present tense for general statements and interpretations.”

In summary, the manuscript shows significant improvement and effectively addresses most prior concerns. The analysis is robust, supported by thorough sensitivity checks. However, by addressing the major suggestions mentioned above, including a White British group in PRS evaluation, accounting for ancestry-socioeconomic confounding, and validating findings in external cohorts would further strengthen the study. Minor edits on sample description and tense consistency are also recommended.

We thank the reviewer for their thoughtful assessment of our work and for their helpful comments throughout.

Reviewer #3 (Remarks to the Author):

We thank the reviewer for their thoughtful co-review of our work.

Reviewer #4 (Remarks to the Author):

All of my concerns have been well addressed.

We thank the reviewer for their thoughtful assessment and helpful comments on the earlier version.

Review for “Three Open Questions in Polygenic Score Portability”

Joyce Y. Wang et al described three open questions for PGS analysis, including high variability, trait-specific portability and measurement impact. Overall, the study highlights that understanding PGS portability requires considering more than just global ancestry, including trait evolution, genetic architecture, social factors, and PGS construction. This approach could lead to more effective and equitable genomic research.

The study topics are very interesting. Manuscripts are well-written with clear illustration. Detailed major comments can be found in the following section.

Major:

1. A “table 1” should be included to describing the distribution of data set. Since authors examined variables like Townsend Deprivation Index, Household Income etc, it would be necessary to compare between GWAS sample and PGS sample to make sure there is no significant differences between the group. Besides variables used in the analysis, some other variables should be included in table 1 include age, sex, location (England, Scotland or Wales).
2. Line 99-100 & Section Prediction accuracy is weakly predicted by genetic distance & Figure 2: Authors split samples into a bin of approximately 260 individuals and compared the prediction accuracy against the genetic distance from the GWAS sample. (1) In Figure 2B regarding weight, the smoothed curve around 100 on x-axis seems to be influenced by a few points on the top. Some further analysis could be done to rule out the randomness of the sample split. For example, would that be possible to vary the number of bins (currently 500) to see if the current trend would maintain. (2) Additionally, currently analysis uses cubic spline fits with 8 knots to capture the trajectory of the trend. It would be helpful to fit some models, like GAM (generalized additive model) to show if the trend is significant or not. And adding the CI (confidence interval) around the smoothed line.
3. Line 183-189 in Discussions: Authors observed some trait-specific trends in portability. It would be great to have some additional analysis to examine what groups of traits might need additional inspections.
4. Line 287-297: Authors detailed described how GWAS are constructed within UKB. (1) Did authors also take sample relatedness into consideration? Either model the sample relatedness or excluded those individuals that are closely related. These individuals might influence GWAS performance. Some more recent software, like REGENIE would be a better approach for modeling GWAS. (2) UKB data is collected at 3 regions, ie England, Scotland and Wales, in

multiple centers in different batch. It would be great to also control assessment center (field id 54) and genotype array (field 22000) in the analysis.

5. Line 298-300: Authors use clumping and thresholding method to construct PGS model. Would that be possible to utilize other methods, like LDpred2 and PRSCS to examine the performance of PGS? Literatures have showed that these methods show a better performance comparing to clumping and thresholding strategy.

Minor:

1. Figure 1C: would that be possible to have a similar histogram for UKB samples?
2. It would be interesting to examine relationship between genetic PCs and F_{st} estimated in the study.
3. Line 171-176 & Discussions: It would be great to examine some admixed ancestry population performance considering the mosaic genetic structure. F_{st} is still a measure of genetic distance in a global level.
4. Line 340: field should be 22189 instead of 189.